# Mixed Nash for Robust Federated Learning

**Wanyun Xie**                                                                                  *wanyun.xie@epfl.ch*
*Laboratory for Information and Inference Systems (LIONS), EPFL*

**Thomas Pethick**                                                                            *thomas.pethick@epfl.ch*
*Laboratory for Information and Inference Systems (LIONS), EPFL*

**Ali Ramezani-Kebrya**                                                                                *ali@uio.no*
*Department of Informatics, University of Oslo and Visual Intelligence Centre*
*Integreat, Norwegian Centre for Knowledge-driven Machine Learning*

**Volkan Cevher**                                                                              *volkan.cevher@epfl.ch*
*Laboratory for Information and Inference Systems (LIONS), EPFL*

**Reviewed on OpenReview:** https://openreview.net/forum?id=mqMzerrVOB

## Abstract

We study *robust* federated learning (FL) within a game theoretic framework to alleviate the server vulnerabilities to even an informed adversary who can tailor training-time attacks (Fang et al., 2020; Xie et al., 2020a; Ozfatura et al., 2022; Rodríguez-Barroso et al., 2023). Specifically, we introduce RobustTailor, a simulation-based framework that *prevents the adversary from being omniscient* and derives its convergence guarantees. RobustTailor improves robustness to training-time attacks significantly with a minor trade-off of privacy. Empirical results under challenging attacks show that RobustTailor performs close to an *upper bound* with perfect knowledge of honest clients.

## 1 Introduction

Federated learning (FL) distributes model training across multiple client devices for improved learning performance with privacy (Konečnỳ et al., 2016; McMahan et al., 2017; Kairouz et al., 2021). However, this distributed nature introduces vulnerabilities, including the risk of attacks from adversarial clients, like Byzantine clients who can send malicious messages instead of correct gradients (Kairouz et al., 2021). Ensuring robustness against such adversaries is critical.

Traditionally, Byzantine-resilience in FL relies on median-based aggregation methods like Krum (Blanchard et al., 2017), Coordinate-wise Median (Comed) (Yin et al., 2018), and Trimmed Mean (TM) (Yin et al., 2018), known for their resilience against random attacks. Nevertheless, these methods may not withstand tailored attacks from powerful adversaries who can exploit knowledge of the aggregation rule (Fang et al., 2020; Xie et al., 2020a; Ozfatura et al., 2022; Rodríguez-Barroso et al., 2023; Baruch et al., 2019). Achieving universal immunity to all attacks is a significant challenge due to the adversary's information leverage.

To partially address this challenge, we formulate a robust distributed learning problem against training-time attacks as a game between a server and an adversary. To improve robustness against the adversary and prevent the adversary from being omniscient, we use a *mixed strategy*, where the server's action set includes a number of robust aggregation rules and where the adversary's action set features a set of attack algorithms. We then propose RobustTailor, a scheme based on simulating aggregation rules under different attacks. To diminish the attacker's information leverage, we assume that the server has access to a *rough estimate* of true gradient update from a public dataset with *a small amount of data*. We theoretically and empirically address scenarios where the computed gradient using public dataset is only a rough estimate of the actual true gradient.

Formulating the game between the server and adversary is challenging as the informed adversary has *an advantage* over the server and can perfectly estimate the empirical mean of all honest updates without an attack, *i.e.,* the ideal update under no attack; while the server does not know the set of honest clients and does not have such prior knowledge of the ideal update. To control the information gap between the server and adversary, we propose to simulate the hypothetical minimax problem through generating simulated gradients, constructing a surrogate loss for the server, and formulating a MixedNash problem that optimizes the probability distributions over attack algorithms and aggregation rules. Solving the MixedNash problem efficiently is another major challenge. Leveraging on the noisy nature of stochastic gradients, we consider a bandit feedback model with limited feedback for RobustTailor, in which the server and adversary can only observe the loss through exploration.

Our theory guarantees convergence of RobustTailor if the number of simulation rounds and the number of honest clients are sufficiently large. Our empirical results demonstrate that RobustTailor provides high resilience to training-time attacks while maintaining stable performance even under a challenging new mixed attack strategy. We emphasize that RobustTailor is expandable, i.e., any Byzantine-resilient scheme can be added in server's aggregation pool and similarly any attack can be considered. Finally, we also provide extensive experiments to show the performance of RobustTailor under unknown attacks out of the server's expectation, poisoned data mixed in the public dataset, subsampling in FL, dynamic strategy of adversary, and the adversary with partial knowledge.

**Summary of Contributions**

- We frame robust distributed learning problem as *a game* between a server and an adversary. We propose RobustTailor, a framework simulating two players using a *bandit feedback model* to improve robustness of FL by choosing a suitable aggregator from existing aggregation rules for each update.

- Theoretically, we establish convergence guarantees for RobustTailor. Our theory implies that RobustTailor converges if the server can roughly estimate the honest update and the number of simulation rounds and honest clients is sufficiently large.

- Empirically, we show that RobustTailor outperforms known aggregation rules and remains robust even under realistic scenarios including mixed attack strategy, unknown attacks out of the server's expectation, poisoned data mixed in the public dataset, subsampling in FL, dynamic strategy of adversary, and the adversary with partial knowledge.

## 2  Related Work

In this section, we provide a summary of related work. See Appendix A for a more complete treatment.

**Training-time attacks in FL.** FL suffers from training-time attacks (Biggio et al., 2012; Bhagoji et al., 2019; Demontis et al., 2019), which potentially participate in every training round and even are adaptive according to leveraging on the knowledge of the structure of aggregation rule. In *model poisoning*, a class of training-time attacks, an adversary controls some clients and directly manipulates their outputs aiming to bias the global model towards the opposite direction of honest training (Kairouz et al., 2021). If Byzantine clients have access to the updates of honest clients, they can tailor their attacks and make them difficult to detect (Fang et al., 2020; Xie et al., 2020a).

**Robust aggregation and Byzantine resilience.** To improve robustness under general Byzantine clients, a number of robust aggregation schemes have been proposed, which are mainly inspired by robust statistics such as median-based aggregators (Yin et al., 2018; Chen et al., 2017), Krum (Blanchard et al., 2017), trimmed mean (TM) (Yin et al., 2018). Moreover, Cao & Lai (2019); Fang et al. (2020); Cao et al. (2021); Xie et al. (2020a) propose server-side verification methods using auxiliary data. ByzantineSGD (Alistarh et al., 2018) and centered clipping (CC) (Karimireddy et al., 2021) are history-aided aggregators. Although these Byzantine-resilient aggregators defend successfully against some attacks, Fang et al. (2020); Xie et al. (2020a); Gouissem et al. (2022) argue that those aggregation rules are vulnerable and fail easily when an informed adversary tailors a careful attack.

Indeed, Krum and Comed are vulnerable to reverse attacks (Fang et al., 2020); TM would be ruined by Alittle attack (Baruch et al., 2019); CC would fail under relocated orthogonal perturbation (ROP) attack (Ozfatura et al., 2022). Rodríguez-Barroso et al. (2023) show that it may not be feasible to design one aggregation rule that is *universally* effective against any attack. Alternatively a pool of existing robust aggregators can be leveraged to establish robustness to a broad range of attacks while no individual aggregator is able to be robust to all of them. Ramezani-Kebrya et al. (2022) propose a framework based on uniform randomization of multiple aggregation rules. Such framework increases complexity of designing an effective attack but cannot promise a suitable aggregator at each training iteration. This paper proposes a framework to select a proper aggregation rule proactively during training.

**Public dataset in FL.** Kairouz et al. (2021, Section 3.1.1) suggest that a small global shared dataset can be used in FL to improve robustness. This dataset may originate from a publicly available proxy data source, a separate dataset from the clients' data that is not privacy sensitive, or perhaps a distillation of the raw data following (Wang et al., 2018). Such public dataset enjoys wide acceptance in FL (Kairouz et al., 2021; Fang & Ye, 2022; Huang et al., 2022; Yoshida et al., 2020). For instance, Sageflow (Park et al., 2021), Zeno (Xie et al., 2019b), and Zeno++ (Xie et al., 2020b) utilize such public data at the server to combat adversarial threats.

The availability of public data at the server also enables collaborative model training with formal differential or hybrid differential privacy guarantees (Gu et al., 2023; Kairouz et al., 2021; Avent et al., 2017). Avent et al. (2017) introduce hybrid differential privacy where some opt-in users voluntarily donate data. Many companies (*e.g.,* Mozilla and Google) rely on a group of testers with higher levels of mutual trust who voluntarily opt-in to a less privacy-preserving model than that of an average end-user.

## 3 Problem Setting

We consider a general distributed system consisting of a parameter server and $n$ clients (Abadi et al., 2016a). Under a synchronous setting in FL, clients compute their updates on their own local data and then aggregate from all peers to update model parameters, and the goal is to solve:

$$\min_{\mathbf{x} \in \mathbb{R}^d} F(\mathbf{x}) = \frac{1}{n} \sum_{i=1}^{n} F_i(\mathbf{x}) \tag{FL}$$

where $F_i : \mathbb{R}^d \to \mathbb{R}$ denotes the training error (empirical risk) of model $\mathbf{x}$ on the local data of client $i$. For example, a popular training algorithm is *stochastic gradient descent*. At iteration $t$, the server sends the global model $\mathbf{x}_t$ to the clients. Each client computes the gradient $\mathbb{E}[\mathbf{g}_i(\mathbf{x}_t)] = \nabla F_i(\mathbf{x}_t)$ on its local training dataset and sends $\nabla F_i(\mathbf{x}_t)$ to the server. The server updates the global model by averaging $\mathbf{x}_{t+1} = \mathbf{x}_t - \frac{1}{n} \sum_{i=1}^{n} \nabla F_i(\mathbf{x}_t)$.

### 3.1 Threat Model and Game Construction

In practical scenarios, clients are vulnerable where powerful adversaries such as informed adversary can control some clients and send malicious updates to the server. Suppose that $f$ Byzantine clients are controlled by an adversary and behave arbitrarily. At iteration $t$, *honest* clients compute and send *honest* stochastic gradients $\mathbb{E}[\mathbf{g}_i(\mathbf{x}_t)] = \nabla F_i(\mathbf{x}_t)$ for $i \in [n-f]$ while *Byzantine* clients, controlled by an adversary, output *attacks* $\mathbf{b}_j \in \mathbb{R}^d$ for $j \in \{n-f+1, \ldots, n\}$. The server receives all $n$ updates and aggregates them following a particular *robust* aggregation rule, which outputs an updated model $\mathbf{x}_{t+1} \in \mathbb{R}^d$. Finally, the server broadcasts $\mathbf{x}_{t+1}$ to all clients. Note that our threat model is common in FL following existing literature (Ramezani-Kebrya et al., 2022; Fang et al., 2020; Cao et al., 2021).

We frame this distributed learning problem under training-time attacks as a game played by the adversary and the server. The adversary aims at corrupting training while the server aims at learning an effective model, which achieves a satisfactory overall empirical risk over honest clients in Eq. (FL). The informed adversary and training-time attacks are described in Section 3.1.1. The server's aggregators are described in

---

Notation can be found prior to the appendix.

Section 3.1.2. To the best of our knowledge, our work is the *first framework* that selects suitable aggregators actively by framing robust learning problem under training-time tailored attacks as a game.

### 3.1.1 Informed adversary with attacks

**Adversary's goal:** Following the training-time attacks literature (Fang et al., 2020; Xie et al., 2020a; Biggio et al., 2012; Bhagoji et al., 2019; Demontis et al., 2019), the adversary's objective is to manipulate the global model at the server and minimize test accuracy *e.g.,* in the classification task.

**Assumption 1** (Informed adversary). *1) An informed adversary controls $f$ out of $n$ clients where these colluding Byzantine clients aim at disturbing the entire training process by sending training-time attacks (Biggio et al., 2012; Bhagoji et al., 2019; Demontis et al., 2019); 2) The number of Byzantine clients is bounded $2f+1 \leq n$ (Blanchard et al., 2017; Alistarh et al., 2018; Ramezani-Kebrya et al., 2022; Karimireddy et al., 2022); otherwise the adversary will be able to provably control the optimization trajectory and set the global model arbitrarily (Lamport et al., 1982); 3) The informed adversary has full knowledge of the outputs of $n-f$ honest clients and controls the outputs of those compromised clients, e.g., their gradients across the course of training.*

Due to having access to the gradients of honest nodes, the adversary can compute the global aggregated gradient of an omniscient aggregation rule, which is the empirical mean of *all honest updates*:

$$\mathbf{g}^* = \frac{1}{n-f} \sum_{i=1}^{n-f} \mathbf{g}_i. \tag{1}$$

**Definition 1** (Attack algorithm). *Let $\{\mathbf{g}_1, \ldots, \mathbf{g}_{n-f}\}$ denote the set of honest updates computed by $n-f$ honest clients. The adversary designs $f$ Byzantine updates using an AT algorithm:*

$$\{\mathbf{b}_{n-f+1}, \ldots, \mathbf{b}_n\} := \text{AT}(\mathbf{g}_1, \ldots, \mathbf{g}_{n-f}, \mathcal{A}) \tag{2}$$

*where $\mathcal{A}$ denotes the set of aggregators formally defined in Section 3.1.2.*

**Adversary's capability:** When the adversary knows a particular server's robust aggregation rule, it is able to design tailored attacks using $n-f$ honest gradients (Fang et al., 2020). We suppose that the adversary has a set of $S$ computationally tractable algorithms to design tailored attacks:

$$\mathcal{F} = \{\text{AT}_1, \text{AT}_2, \ldots, \text{AT}_S\}. \tag{3}$$

It is shown that several efficient and tailored attacks can be designed that provably fail SOTA robust aggregators, *e.g.,* Krum, Comed, TM, and CC (Fang et al., 2020; Xie et al., 2020a; Ozfatura et al., 2022; Rodríguez-Barroso et al., 2023; Baruch et al., 2019).

### 3.1.2 Server with aggregators

**Server's goal:** The server aims at learning an effective model, which achieves a satisfactory overall empirical risk over honest clients comparable to that *under no attack*. To update the global model, the server aggregates all gradients sent by clients at each iteration.

**Definition 2** (Aggregation rule). *Let $\mathbf{g}'_i \in \mathbb{R}^d$ denote an update received from client $i$, which can be either an honest or compromised client for $i \in [n]$. That means $\{\mathbf{g}'_i\}_{i=1}^n = \{\{\mathbf{g}_i\}_{i=1}^{n-f}, \{\mathbf{b}_i\}_{i=n-f+1}^n\}$. The server aggregates all updates from $n$ clients and outputs a global update $\mathbf{g} \in \mathbb{R}^d$ using an aggregation rule AG:*

$$\mathbf{g} = \text{AG}(\mathbf{g}'_1, \ldots, \mathbf{g}'_n, \mathcal{F}) \tag{4}$$

*where $\mathcal{F}$ denotes the set of attacks defined in Section 3.1.1.*

**Assumption 2** (Server). *The server knows the number of compromised clients $f$ or an upper bound on $f$, which is a common assumption in robust learning (Blanchard et al., 2017; Alistarh et al., 2018; Ramezani-Kebrya et al., 2022; Karimireddy et al., 2022; Rajput et al., 2019). However, it does not know the specific Byzantine clients among $n$ clients in this distributed system such that the server cannot compute $\mathbf{g}^*$ in Eq. (1) directly.*

To learn and establish some level of robustness against training-time attacks, several Byzantine-resilient aggregation rules have been proposed, *e.g.,* Krum (Blanchard et al., 2017) and Comed (Yin et al., 2018). These median-based schemes have been shown to be vulnerable to tailored attacks (Fang et al., 2020; Xie et al., 2020a; Karimireddy et al., 2021).

**Server's capability:** The server aggregates all updates from clients by an aggregation rule to update the global model. The set of $M$ aggregators used by the server is denoted by

$$\mathcal{A} = \{\mathrm{AG}_1, \mathrm{AG}_2, \ldots, \mathrm{AG}_M\}. \tag{5}$$

**Assumption 3** (Sets of aggregators and attacks). *We assume that the pool of aggregators $\mathcal{A}$ and the set of attacks $\mathcal{F}$ are known by the server and the adversary. However, the specific $\mathrm{AT}^t$ and $\mathrm{AG}^t$ chosen at iteration $t$ are unknown for each other. To avoid trivial solutions, we assume each aggregation rule is robust (formal definition of robustness is provided in Appendix B) against a subset of attack algorithms in $\mathcal{F}$ while no aggregation rule is immune to all attack algorithms.*

## 3.2 Problem Formulation

To evaluate the performance of an updated global model, *i.e.,* the output of AG in Eq. (4), we define a loss function, which measures the discrepancy of the output of AG and an omniscient model update *obtaining all honest gradients.*

**Definition 3** (Loss function). *The loss function of using aggregation rule* AG *under attack* AT *is defined as*

$$\ell(\mathrm{AG}, \mathrm{AT}, \{\mathbf{g}_i'\}_{i=1}^n) = \|\mathrm{AG}(\{\mathbf{g}_i'\}_{i=1}^n, \mathcal{A}) - \mathbf{g}^*\| = \|\mathrm{AG}(\{\mathbf{g}_i\}_{i=1}^{n-f}, \mathrm{AT}(\{\mathbf{g}_i\}_{i=1}^{n-f}, \mathcal{F}), \mathcal{A}) - \mathbf{g}^*\| \tag{6}$$

*where $\mathbf{g}^*$ is the ideal model under no attack which is computed in Eq. (1).*

To train the global model, the server takes multiple rounds of stochastic gradient descent by aggregating the stochastic gradients from clients. However, some gradients might be corrupt at each round, which are sent by compromised clients controlled by the adversary. We frame this robust distributed learning scenario as a *game* between the adversary and the server. The server aims to minimize the loss defined in Definition 3, while the adversary aims to maximize it. This game as a *minimax problem* is formulated as:

$$\min_{\mathrm{AG} \in \mathcal{A}} \max_{\mathrm{AT} \in \mathcal{F}} \ell(\mathrm{AG}, \mathrm{AT}, \{\mathbf{g}_i'\}_{i=1}^n). \tag{MinMax}$$

Ideally, the game in MinMax reaches a Nash equilibrium (NE) (Nash, 1950). The hypothetical process of model aggregation is shown in Appendix C. However, the server cannot compute the loss since it cannot distinguish honest gradients and does not know $\mathbf{g}^*$. We propose to *simulate* the game in the Section 4.

## 4 Robust Aggregation

Because MinMax cannot be solved during the process of updating the model, we propose to *simulate* it instead and obtain an optimized aggregator for model updates. As mentioned in Section 3, the informed adversary has *an advantage* over the server since it can perfectly estimate $\mathbf{g}^*$ in Eq. (1), while the server does not have such knowledge and cannot identify honest clients a priori. Without any estimate of the true update, the server cannot do better than selecting one of the aggregation rules uniformly at random (Ramezani-Kebrya et al., 2022). This random strategy may output a highly suboptimal aggregation rule and does not leverage the knowledge of $\mathcal{F}$. We assume that the server has access to a *rough estimate* of true gradient update which could control the *information* gap between the server and adversary. Note that we discuss estimated gradients specifically in Section 4.1. Let $\widetilde{\mathbf{g}}$ computed by the server denote the rough estimate of $\mathbf{g}^*$.

**Remark 1** (Rough estimate of $\mathbf{g}^\star$). *Rough estimate of $\mathbf{g}^\star$ for the server can be obtained from 1) a public dataset and 2) gradients of trusted clients. To guarantee convergence, we only require that the update from the public dataset is a **rough estimate** of the ideal $\mathbf{g}^\star$. Through theoretical analysis (Section 5) and experiments (Section 6), we address scenarios where the computed gradient is only a rough estimate of the actual true*

*gradient e.g., by considering distribution shifts and mixture with poisoned data. Note that we do not directly update the model using $\widetilde{\mathbf{g}}$ because 1) we aim to avoid wasting the computation of all clients, and 2) the global model should reflect the distribution of all honest clients not just a small subset.*

For the simulation, the server generates the simulated gradients $\{\widetilde{\mathbf{g}}_i\}_{i=1}^{n-f}$ based on the public dataset or gradients of trusted clients. The server's surrogate loss function in the simulated game is given by

$$\widetilde{\ell}(\mathrm{AG}, \mathrm{AT}, \{\widetilde{\mathbf{g}}_i\}_{i=1}^{n-f}) = ||\mathrm{AG}(\{\widetilde{\mathbf{g}}_i\}_{i=1}^{n-f}, \mathrm{AT}(\{\widetilde{\mathbf{g}}_i\}_{i=1}^{n-f}, \mathcal{A}), \mathcal{F}) - \widetilde{\mathbf{g}}^{\star}||^2 \tag{7}$$

where $\widetilde{\mathbf{g}}^{\star} = \frac{1}{n-f} \sum_{i=1}^{n-f} \widetilde{\mathbf{g}}_i$.

Let $\tilde{\mathbf{L}} \in \mathbb{R}_+^{M \times S}$ denote the surrogate loss of $M$ aggregators corresponding to $S$ attacks, and $\tilde{\mathbf{L}}(\mathrm{AG}_i, \mathrm{AT}_j)$ represents the loss associated with aggregation rule $i$ in $\mathcal{A}$ under attack $j$ in $\mathcal{F}$ in the simulation. After the adversary has committed to a probability distribution $\mathbf{q}$ over $S$ attack algorithms, the server chooses a probability distribution $\mathbf{p}$ over $M$ aggregation rules. Then, the server incurs the loss $\widetilde{\ell}(\mathbf{p}, \mathbf{q}) = \mathbf{p}^{\top} \tilde{\mathbf{L}} \mathbf{q}$. We solve MixedNash below instead of MinMax.

$$\min_{\mathbf{p} \in \Delta_M} \max_{\mathbf{q} \in \Delta_S} \mathbf{p}^{\top} \tilde{\mathbf{L}} \mathbf{q} \tag{MixedNash}$$

where $\Delta_M$ and $\Delta_S$ denote the probability simplex in $[M]$ and $[S]$, respectively.

In practice, it is computationally expensive to compute $\tilde{\mathbf{L}} \in \mathbb{R}_+^{M \times S}$. Additionally, stochastic gradients are noisy. Therefore, we consider the *bandit feedback model*, in which the server and adversary never observe $\tilde{\mathbf{L}}$ in its entirety, but instead only the loss associated with a particular realization of play. To solve MixedNash in the bandit feedback model, one player could implement the well-known *Exponential-weight Algorithm for Exploration and Exploitation (Exp3)* (Seldin et al., 2013) whose detailed description is deferred to Appendix D.

We propose an algorithm in which two players (the server and the adversary) in our model simultaneously execute Exp3. We term our proposed *robust aggregation scheme* as RobustTailor, outputting an optimized AG at each iteration. The specific steps from the server's perspective are shown in Algorithm 1. Using the public dataset, the server generates $n - f$ noisy stochastic gradients $\widetilde{\mathbf{g}}_i$ for $i \in [n-f]$. The server also assigns equal initial weights $w^0(i)$ for $i \in [M]$, $v^0(j)$ for $j \in [S]$ to all aggregators and attacks independently. In each round, it chooses an aggregator and an attack based on probability distributions determined by the current weights. Then, it observes the loss and update the weights based on the received regret/reward associated with the chosen aggregator/attack, adjusting more for poorly performing actions and less for well-performing ones. After $K$ rounds of simulation on $\{\widetilde{\mathbf{g}}_i\}_{i=1}^{n-f}$, the server obtains a final probability distribution $\mathbf{p}$ and selects an aggregation rule by sampling from $\mathbf{p}$. The steps for our robust training procedure are summarized in Algorithm 2.

The adversary can also perform simulation to *optimize* its attack at each iteration. The main differences for an adversarial simulation compared to RobustTailor include: 1) the adversary can use perfect honest stochastic gradients $\{\mathbf{g}_i\}_{i=1}^{n-f}$ instead of noisy estimates; 2) the probability output is $\mathbf{q}$ which is calculated by the weight vector of attacks $v(j)$ for $j \in [S]$. The details of the adversarial simulation are provided in Appendix E.

## 4.1 Privacy

Given the importance of privacy in FL, this is a deliberate trade-off between enhancing robustness and preserving privacy. We propose two scenarios to obtain estimated gradients $\tilde{\mathbf{g}}$ in a privacy-preserving manner. In the first scenario under **less privacy concerns**, the server has a public dataset consisting of a small amount of data donated by clients or a trusted party. The existence of such public dataset is a valid and common assumption in FL (Kairouz et al., 2021).

In another scenario with strict privacy requirements, the server trusts only a small subset of honest clients providing reliable updates. It is common for the server to have a core group of trustworthy clients, like companies relying on highly trusted testers. Notably, RobustTailor is compatible with *all privacy-preserving techniques*, such as differential privacy (DP) (Bassily et al., 2014; Wei et al., 2020). Further details are provided in Appendix I. In both scenarios, RobustTailor does not significantly compromise privacy and is indeed acceptable given its benefits as shown in Section 6.4.

---

**Algorithm 1** RobustTailor

---

**Input:** Updating rates $\lambda_1$, $\lambda_2$, $\tilde{\lambda}_1$ and $\tilde{\lambda}_2$, simulation rounds $K$, simulated gradients $\{\widetilde{\mathbf{g}}_i\}_{i=1}^{n-f}$, $\mathcal{A}$, $\mathcal{F}$.
Initialize weight vector $w^0(i) = 1$ for $i \in [M]$ and $v^0(j) = 1$ for $j \in [S]$.
**for** $k = 1$ **to** $K$ **do**

> Set $\mathbf{p}^k(\widetilde{\mathrm{AG}_i}) = (1 - \lambda_1) \frac{w^k(i)}{\sum_{i=1}^{M} w^k(i)} + \lambda_1 \frac{1}{M}$ for $i \in [M]$.
>
> Set $\mathbf{q}^k(\widetilde{\mathrm{AT}_j}) = (1 - \lambda_2) \frac{v^k(j)}{\sum_{j=1}^{S} v^k(j)} + \lambda_2 \frac{1}{S}$ for $j \in [S]$.
>
> Sample $\mathrm{AG}^k \sim \mathbf{p}^k$ and $\mathrm{AT}^k \sim \mathbf{q}^k$ respectively.
> Estimate the loss $\ell^k = \widetilde{\ell}(\mathrm{AG}^k, \mathrm{AT}^k, \{\widetilde{\mathbf{g}}_i\}_{i=1}^{n-f})$.
> For $i \in [M]$, set $\hat{\ell}_1^k(i) = \frac{\mathbb{I}\{\widetilde{\mathrm{AG}_i} = \mathrm{AG}^k\}}{\mathbf{p}^k(\widetilde{\mathrm{AG}_i})} \ell^k$,    $w^{k+1}(i) = w^k(i) \exp(-\tilde{\lambda}_1 \hat{\ell}_1^k(i)/M)$.
> For $j \in [S]$, set $\hat{\ell}_2^k(j) = \frac{\mathbb{I}\{\widetilde{\mathrm{AT}_j} = \mathrm{AT}^k\}}{\mathbf{q}^k(\widetilde{\mathrm{AT}_j})} \ell^k$,    $v^{k+1}(j) = v^k(j) \exp(\tilde{\lambda}_2 \hat{\ell}_2^k(j)/S)$.

Set $\mathbf{p}_i = \frac{\sum_{k=1}^{K} \mathbf{p}^k(\widetilde{\mathrm{AG}_i})}{K}$ for $i \in [M]$.
Sample $\mathrm{AG} \sim \mathbf{p}$.
**Output:** AG.

---

**Algorithm 2** Server's aggregation

---

**Input:** Learning rate $\eta_t$, $n$ clients, $f$ compromised clients, iteration rounds $T$, $\mathcal{A}$ and $\mathcal{F}$
Initialize model $\mathbf{x}_0$.
**for** $t = 1$ **to** $T$ **do**

> Send $\mathbf{x}_t$ to all clients.
> Receive gradients from all clients $\{\mathbf{g}_i'\}_{i=1}^{n}$.
> Calculate simulated gradients $\{\widetilde{\mathbf{g}}_i\}_{i=1}^{n-f}$.
> Call Algorithm 1 to aggregate $\mathrm{AG}^t = \mathrm{RobustTailor}(\{\widetilde{\mathbf{g}}_i\}_{i=1}^{n-f}, \mathcal{A}, \mathcal{F})$.
> Update the global model by $\mathbf{x}_{t+1} = \mathbf{x}_t - \eta_t \mathrm{AG}^t(\{\mathbf{g}_i'\}_{i=1}^{n})..$

---

### 4.2 Computational complexity

Appendix K demonstrates both theoretical analysis and empirical results of RobustTailor's computation complexity. Our fine-grained analysis shows that the overall time complexity of RobustTailor is given by $\mathcal{O}((M + S)\overline{T}K)$ where the average complexity per round $\overline{T}$ depends on the aggregators in server's pool.

## 5 Theoretical Guarantees

To provide guarantees of the outer loop in Algorithm 2, we first show convergence of the inner optimization described in Algorithm 1. Any two simultaneously played no-regret algorithms for a minimax problem can be turned into convergence to a NE (see Lemma 2 in Appendix F). Combined with no-regret properties of Exp3, we obtain guarantees for the aggregation rule returned from Algorithm 1. Considering the information asymmetry involved in Algorithm 1, $L$ is replaced by the simulation loss $\widetilde{\ell}$ shown in Eq. (7).

**Lemma 1** (Bounded loss of simulated game). *Let $\widetilde{\ell}$ be the simulation loss in Eq. (7). Sample $\mathrm{AG} \sim \mathbf{p}$ as defined in Algorithm 1 with $\tilde{\lambda}_1 = \sqrt{\frac{\log M}{KM}}$ and $\tilde{\lambda}_2 = \sqrt{\frac{\log S}{KS}}$. Then the loss is bounded in expectation for any attack $\mathrm{AT} \in \mathcal{F}$:*

$$\mathbb{E}_{\mathrm{AG}}\left[\widetilde{\ell}\left(\mathrm{AG}, \mathrm{AT}, \{\widetilde{\mathbf{g}}_i\}_{i=1}^{n-f}\right)\right] \leq \mathbf{p}^{\star\top} \tilde{\mathbf{L}} \mathbf{q}^\star + 2\frac{\sqrt{M \log M} + \sqrt{S \log S}}{\sqrt{K}}, \tag{8}$$

*where $(\mathbf{p}^\star, \mathbf{q}^\star) \in \Delta_M \times \Delta_S$ is a Nash equilibrium of the zero-sum game with the payoff matrix $\tilde{\mathbf{L}}$ as defined in MixedNash.*

Lemma 1 implies that the simulated loss approaches the NE value even under the worst-case attack. Note that Lemma 1 gives the guarantees for *inner* loop only considering simulated gradients. Now we extend it to *outer* loop whose ideal update is $\mathbf{g}^\star$.

We assume the standard assumption of bounded estimation error through decomposing the error into bias and variance terms.

**Assumption 4** (Bounded estimation error). *Let $i \in [n - f]$. The simulated gradient $\widetilde{\mathbf{g}}_i$ has a bounded estimation error $\mathbb{E}[\|\widetilde{\mathbf{g}}_i - \mathbf{g}^\star\|^2] \leq \mathsf{B}_{\mathrm{est}} + \mathsf{V}_{\mathrm{est}}$ where the bias and variance bounds are given by:*

$$\|\mathbb{E}[\widetilde{\mathbf{g}}_i] - \mathbf{g}^\star\|^2 \leq \mathsf{B}_{\mathrm{est}}, \quad \mathbb{E}[\|\widetilde{\mathbf{g}}_i - \mathbb{E}[\widetilde{\mathbf{g}}_i]\|^2] \leq \mathsf{V}_{\mathrm{est}}.$$

**Theorem 1** (Outer loop's convergence). *Under Assumption 4, suppose $\{\eta_t\}_{t=1}^\infty$ in Algorithm 2 satisfies $\sum_t \eta_t = \infty$ and $\sum_t \eta_t^2 < \infty$. Let $\mathbf{x}_t$ denote the output of AG in Algorithm 2 with bounded norm $C$ for $t \geq 1$. For a nonconvex loss function $F$ in Eq. (FL), which is three times differentiable with continuous derivatives, bounded from below, and satisfies (Bottou, 1998, Assumption iv in Section 5.1), we have $\nabla F(\mathbf{x}_t) \to 0$ a.s. if*

$$\mathbb{E}\|\widetilde{\mathbf{g}}^\star\|^2 \geq \frac{C\mathsf{V}_{\mathrm{est}}}{n-f} + \mathbb{E}[\mathbf{p}^{\star\top}\tilde{\mathbf{L}}\mathbf{q}^\star] + C\mathsf{B}_{\mathrm{est}} + 2\frac{\sqrt{M \log M} + \sqrt{S \log S}}{\sqrt{K}} \tag{9}$$

*where the expectation is over simulated gradients.*

The condition in Theorem 1 is satisfied when the number of simulation rounds $K$ and the number of honest clients $n - f$ are sufficiently large. The optimal strategy $\mathbf{p}^\star$ depends on the realization of the gradients, so $\mathbb{E}[\mathbf{p}^{\star\top}\tilde{\mathbf{L}}\mathbf{q}^\star]$ is small as long as there exists an effective aggregation rule for each realization. Furthermore, $\mathsf{B}_{\mathrm{est}}$ is small when the public dataset represents the clients' data distribution by some extent. If $\mathsf{B}_{\mathrm{est}}$ is too large, it may be the case that the worst $\mathrm{AG} \in \mathcal{A}$ is selected due to the mismatch with the true updates.

To the best of our knowledge, this is the first convergence guarantee on nonconvex, heterogeneous, and simulated setting where the adversary and the server play an asymmetrical game to optimize their strategies.

The proofs of Lemma 1 and Theorem 1 are provided in Appendices F to H.

## 6 Experimental Evaluation

In this section, we evaluate the resilience of RobustTailor against tailored attacks. To provide both intuitive results showing robustness of RobustTailor and its realistic implementation in FL, we first construct a basic setting and then extend it to various scenarios. In *the basic setting*, we simulate training with a total of 12 clients, 2 of which are compromised by an informed adversary. We train a CNN model on MNIST (Lecun et al., 1998) under independent and identically distributed (iid) setting. For the server, we construct a simple pool of aggregators including only Krum (Blanchard et al., 2017) and Comed (Yin et al., 2018). For the adversary, we consider two tailored attacks which can successfully ruin Krum and Comed respectively and and two stronger mixed attacks. Referring to (Fang et al., 2020; Xie et al., 2020a), $\epsilon$-*reverse attack* with a small $\epsilon$ corrupts Krum while a large $\epsilon$ corrupts Comed. We choose $\epsilon = 0.5$ and $\epsilon = 100$ as the basic set of attacks. Note that $\epsilon$-reverse attack submits the scaled honest update with a parameter $-\epsilon$ to the server. To enable simulation, we assume honest clients donate 5% of their local training data to the server as a public dataset elaborated in Remark 1. The informed adversary has access to the gradients of honest clients. Note that all experiments without specific clarification follow the basic setting, and the details of the model and training hyper-parameters are provided in Appendix J.1.

For extensive experiments, we train the CNN model on Fashion-MNIST (FMNIST) (Xiao et al., 2017) and CIFAR10 (Krizhevsky & Hinton, 2009) datasets. We also consider non-iid settings with different heterogeneous degrees. For the pool of the server's aggregation rules, we further consider SOTA aggregators (TM (Yin et al., 2018), Geomed (GM) (Pillutla et al., 2022)), mixed strategy (Bulyan (El Mhamdi et al., 2018)), history-aided aggregator (CC (Karimireddy et al., 2021)), server-side verification methods (ERR (Fang et al., 2020) and LFR(Fang et al., 2020)). Regarding attacks, we propose a simulation-based attack strategy AttackTailor, and implement model poisoning attacks (Mimic (Karimireddy et al., 2022), Alittle (Baruch et al., 2019)), data poisoning attacks (label flipping (LF) (Muñoz-González et al., 2017), and random label (LR) (Zhang et al., 2021)).

## 6.1 Basic Experiments

To validate our theoretical results, we first show the empirical results based on the *basic setting*, which strictly satisfies the assumption that the server/the adversary knows $\mathcal{F}/\mathcal{A}$ but does not know AT/AG at each iteration. We test RobustTailor against an adversary with both single tailored attacks and stronger mixed attacks. Moreover, we show the results in the non-iid settings with three different heterogeneous degrees in Appendix J.2.

**Single tailored attacks.** RobustTailor successfully decreases the capability of the adversary to launch tailored attacks. RobustTailor maintains stability in Fig. 1a when Krum fails catastrophically under a small $\epsilon$ attack. Fig. 1b shows that RobustTailor has much less fluctuations in terms of test accuracy compared to Comed when facing a large $\epsilon$ attack. In addition, on average, RobustTailor has 70.68% probability of choosing Comed under $\epsilon = 0.5$ attack while 65.49% probability of choosing Krum under $\epsilon = 100$ attack, which validates that the server successfully learns how to defend. Training on FMNIST shows consistent results as seen in Fig. 1c and Fig. 1d. Additional results on CIFAR10 are in Appendix J.2.

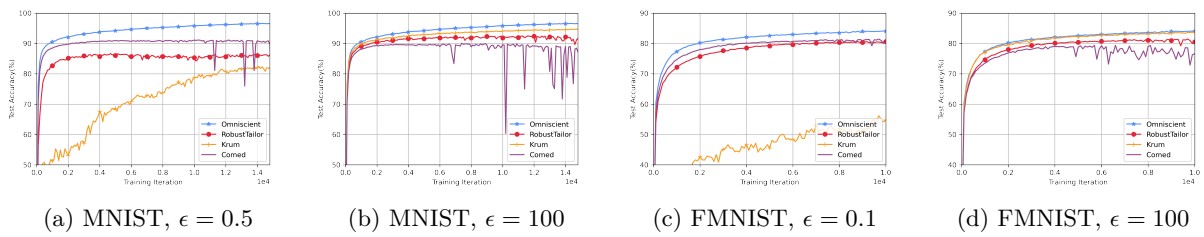

|  (a) MNIST, $\epsilon = 0.5$ | (b) MNIST, $\epsilon = 100$ | (c) FMNIST, $\epsilon = 0.1$ | (d) FMNIST, $\epsilon = 100$ |

Figure 1: **Test accuracy on MNIST and FMNIST under iid setting.** Attacks with $\epsilon \in \{0.1, 0.5, 100\}$ are applied. RobustTailor selects an aggregator from Krum and Comed based on the simulation.

**Mixed attacks.** We now consider two mixed and stronger attack strategies. Building on the basic attack set, **StochasticAttack** shown in Fig. 2a picks an attack from $\epsilon = 0.5$ and $\epsilon = 100$ reverse attacks *uniformly at random* at each iteration. **AttackTailor** in Fig. 2b optimizes an attack based on simulation, whose detailed algorithm is in Appendix E. Compared to all previous attacks including StochasticAttack, AttackTailor is much stronger since it can pick a proper attack under perfect knowledge of honest updates. The poison of AttackTailor is almost as effective as the most targeted attack tailored against a single deterministic aggregator. Importantly, RobustTailor shows impressive robustness when facing such a strong adversary like AttackTailor.

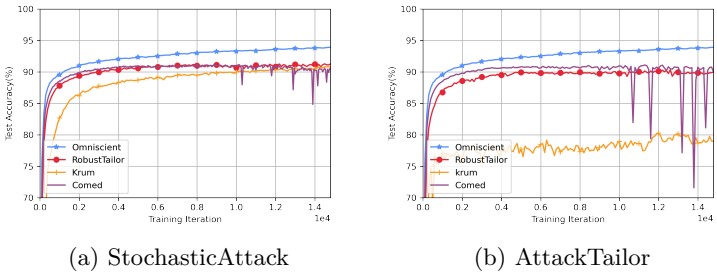

|  (a) StochasticAttack | (b) AttackTailor |

Figure 2: **Mixed attacks.** StochasticAttack applies $\epsilon = 0.5$ and $\epsilon = 100$ uniformly at random. Strong AttackTailor *optimizes* the attack via simulation with perfect knowledge of honest updates.

## 6.2 Dynamic Strategies

In FL, the server may randomly choose a subset of clients for aggregation at each iteration especially when there exists a large number of clients, e.g., in cross-device FL. The adversary could also apply such dynamic strategy by picking a subset of malicious clients to attack at each iteration, which would increase the complexity of attack. In this section, we discuss these two scenarios below.

**Subsampling by the server.** Subsampling, a common technique in large-scale FL, can increase the complexity of attacks. The server picks a subset of clients randomly for updates at each iteration. The adversary can know which clients are selected and leverage the selected honest updates to design the attack for the compromised clients that are chosen. To simulate subsampling in FL with a large number of clients, we run additional experiments with 120 clients, in which 20 Byzantines are compromised by an adversary. The server chooses 10% of clients randomly for aggregation and assumes 2 of 12 clients are compromised at each iteration. We decrease the learning rate from 0.01 to 0.001 because all aggregation methods are too unstable under the original setting. The results shown in Fig. 3 are consistent with the results without subsampling.

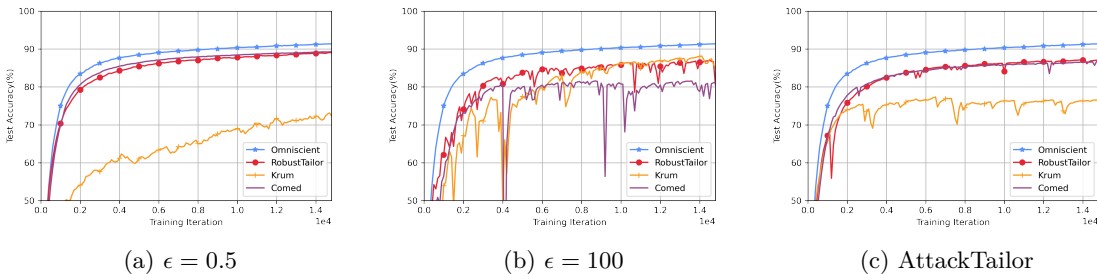

(a) $\epsilon = 0.5$          (b) $\epsilon = 100$          (c) AttackTailor

Figure 3: **Subsampling by the server.**

**Dynamic strategy of the adversary.** The adversary can also use a dynamic strategy by changing the number of malicious updates dynamically. We consider a setting where the adversary picks $1 - 3$ clients randomly to control at each iteration; while the server still considers 2 Byzantines among 12 clients. Fig. 4 shows the results. Compared to the results in Fig. 1 and Fig. 2b, RobustTailor still has a good performance although some aggregation rules in the pool are slightly impacted.

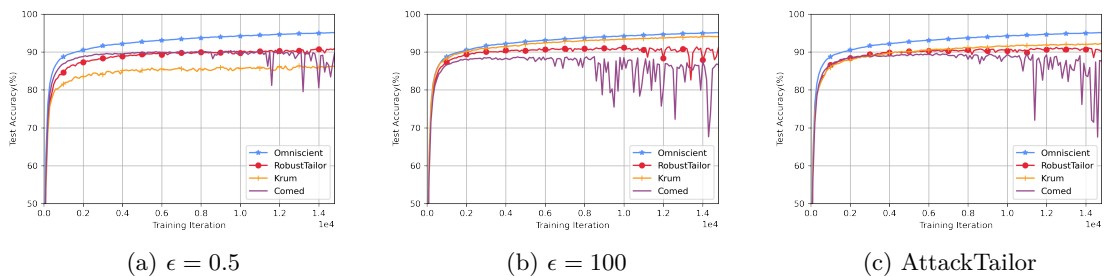

(a) $\epsilon = 0.5$          (b) $\epsilon = 100$          (c) AttackTailor

Figure 4: **Dynamic attack strategy of the adversary.**

## 6.3 Partial Knowledge of Server/Adversary

Considering more realistic scenarios, it is a bit strict assuming server/adversary has the full knowledge of $\mathcal{F}/\mathcal{A}$. In this section, we show the results under partial knowledge of the server and the adversary.

**More aggregators in the server's pool.** RobustTailor is an expandable framework, which can include or replace aggregators in the server's pool. The results in Fig. 5 show that RobustTailor framework improves robustness when more aggregators are added to the basic pool. Additional aggregators added here are TM (Yin et al., 2018), GM (Pillutla et al., 2022), Bulyan (El Mhamdi et al., 2018), and CC (Karimireddy et al., 2021).

**Unknown attacks for the server.** We now address the important question of "What will happen if there is an attack out of the server's expectation?" Fig. 6 shows the results when the server does not know the set of attacks. In particular, $\epsilon = 0.1$ in Fig. 6a and $\epsilon = 150$ in Fig. 6b are *the same type of attacks* as the basic setting; while Mimic (Karimireddy et al., 2022) in Fig. 6c and Alittle (Baruch et al., 2019) in Fig. 6d are *different types of attacks*. Under Alittle and Mimic attacks, we expand the set of RobustTailor with GM (Pillutla et al., 2022) and Bulyan (El Mhamdi et al., 2018) and decrease the learning rate to 0.005. Fig. 6 shows that RobustTailor can defend against not only attacks similar to expected ones but also those that are

unexpected with completely different structures and designs. As a mixed framework, RobustTailor is hard to fail since the adversary hardly designs a tailored attack failing several aggregation rules simultaneously.

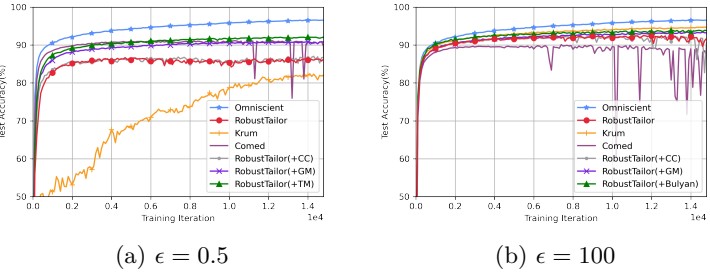

(a) $\epsilon = 0.5$ · (b) $\epsilon = 100$

Figure 5: **More aggregators added to RobustTailor.**

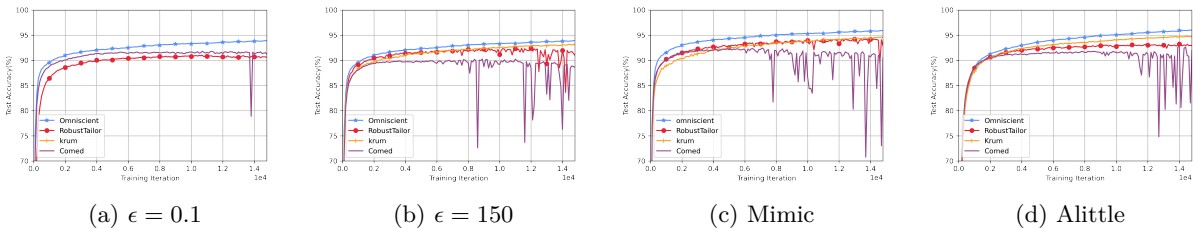

(a) $\epsilon = 0.1$ · (b) $\epsilon = 150$ · (c) Mimic · (d) Alittle

Figure 6: **Unknown attacks for the server.** Performance of Krum under $\epsilon = 0.1$ does not go above 20%.

### 6.4 Imperfect Public Dataset

*Will the quantity and the quality of public dataset impact the performance of* RobustTailor*?* We discuss the size of public dataset and various types of distribution shifts with respect to the original data of honest clients. In particular, we evaluate the robustness of RobustTailor under poisoned data mixed in and data obtained from different data sources. We also show aggregators with auxiliary data used in RobustTailor.

**Size of public dataset.** In practice, it is important to minimize amount of public data as much as possible while maintaining effectiveness of RobustTailor. Therefore, studying the impact of the size of public data is necessary. Fig. 7 shows the performance of RobustTailor with different proportion of public data in the basic setting. It is obvious that the amount of public data has little impact on RobustTailor and even very small proportion of data donated by clients (*e.g.,* 0.1%) helps RobustTailor achieve great performance. These results further validate Remark 1, that only a rough estimate of the actual true update is required using public dataset.

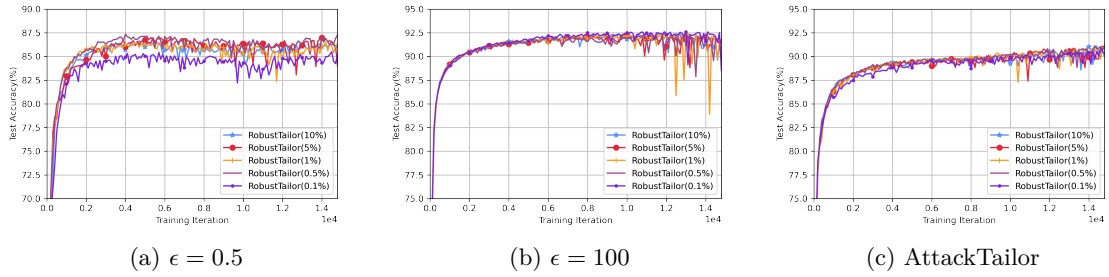

(a) $\epsilon = 0.5$ · (b) $\epsilon = 100$ · (c) AttackTailor

Figure 7: **The impact of the proportion of public data.**

**Poisoned data mixed in the public dataset.** Although most existing aggregators defend data poisoning attacks which are less effective than model poisoning attacks (Kairouz et al., 2021), it is worth considering such attacks because Byzantine clients may be able to donate poisoned data to the public dataset. We assume 16.7%

of data in the public dataset is poisoned due to 16.7% of malicious clients. We consider two data poisoning attacks: LF (Muñoz-González et al., 2017) and LR (Zhang et al., 2021). Fig. 8 demonstrates that poisoned data mixed in the public dataset has little impact on RobustTailor, which also validates that *a small gap* between the public dataset and true samples does not reduce the effectiveness of RobustTailor substantially.

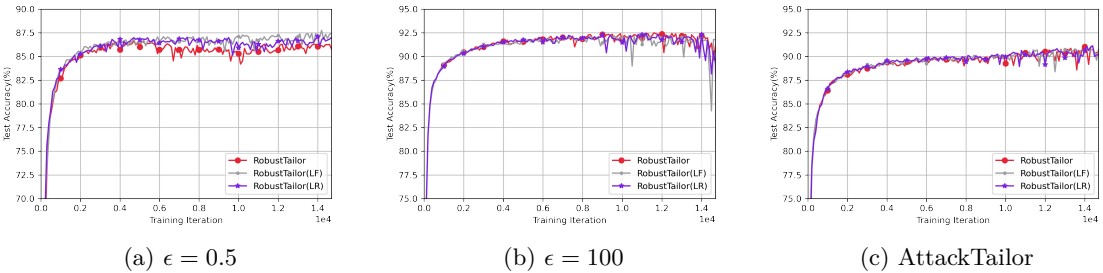

(a) $\epsilon = 0.5$        (b) $\epsilon = 100$        (c) AttackTailor

Figure 8: **Poisoned data mixed in the public dataset.**

**Public data obtained from a trusted party under distribution shifts.** In our paper, we do not require each client to donate data. The public data of the server can also be obtained from a trusted party instead of each client (we discuss this in Remark 1). In the experiments, clients donate a small percentage of data that could be viewed equivalently as a publicly available dataset. We also show experiments that public data helps even when there are challenging and realistic *covariate shifts*, in which we replace the original public dataset with the same number of digits from EMNIST dataset (Cohen et al., 2017). The empirical results shown in Fig. 9 demonstrate that public data helps even under covariate shifts.

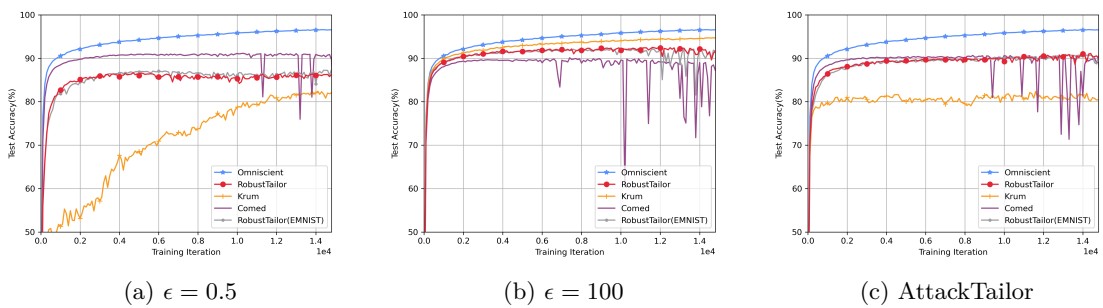

(a) $\epsilon = 0.5$        (b) $\epsilon = 100$        (c) AttackTailor

Figure 9: **Public data obtained from EMNIST.**

**Aggregators with auxiliary data from the public dataset.** Public datasets, in addition to simulation, can aid in aggregation by enabling methods like error rate based rejection (ERR) and loss function based rejection (LFR) (Fang et al., 2020). These methods reject potentially harmful gradients based on error rates or loss values before aggregation. Our experiments with the same setup as in Fig. 2b show that Krum/Comed assisted by ERR/LFR are severely impacted under AttackTailor achieving only **10**% accuracy. In contrast, RobustTailor reaches **90.28**% accuracy, providing further evidence of its superiority over existing techniques. Additional experiments reveal that ERR/LFR helps the baseline aggregator to achieve 97% accuracy under $\epsilon = 0.5$ attack while it is totally ruined under $\epsilon = 100$ attack. In more challenging scenarios where AttackTailor disrupts baseline aggregation rules, RobustTailor continues to perform effectively.

**Discussion.** We have considered scenarios where the server owns a public dataset under distribution shifts in this section. RobustTailor also works for another potential scenario which completely avoids storing any data on the server by simulating with the updates from a small number of trusted clients as long as the rough estimate is provided as discussed in Section 4.1.

**Additional experiments.** To further validate the performance of RobustTailor, we set up additional experiments in Appendix J.2 including 1) three datasets; 2) more Byzantines; 3) non-iid settings; and 4) combination strategies of aggregators.

## 7 Conclusions

We formulate the robust distributed learning problem as a game between a server and an adversary, using RobustTailor to simulate the server's aggregation rules under different adversary attacks. We've established convergence guarantees for RobustTailor. Empirical results highlight RobustTailor's superiority over robust baselines. Furthermore, extensive experiments demonstrate RobustTailor's adaptability and robustness in various realistic scenarios. Additionally, we've investigated the use of imperfect public dataset estimates to enhance FL robustness, showing that RobustTailor with rough ideal update estimates is effective against distribution shifts.

**Acknowledgments**

This work was supported by the Swiss National Science Foundation (SNSF) under grant number 200021_205011. This work was supported by Google. This work was supported by Hasler Foundation Program: Hasler Responsible AI (project number 21043). This work was sponsored by the Army Research Office and was accomplished under Grant Number W911NF-24-1-0048.

The work of Ali Ramezani-Kebrya was supported by the Research Council of Norway through its Centres of Excellence scheme, Integreat - Norwegian Centre for knowledge-driven machine learning, project number 332645; and the Research Council of Norway through its Centre for Research-based Innovation funding scheme (Visual Intelligence under grant no. 309439).

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

**Notation.** We use $\mathbb{E}[\cdot]$, $\|\cdot\|$ to denote the expectation operator, Euclidean norm respectively. We use $|\cdot|$ to denote the length of a binary string, the length of a vector, and cardinality of a set. We use lower-case bold letters to denote vectors. Sets are typeset in a calligraphic font. We use $[n]$ to denote $\{1, \ldots, n\}$ for an integer $n$. We use $\Delta_M$ to denote the probability simplex in $\mathbb{R}^M$.

## A  Complete Related Work

**Federated learning (FL).** FL (Konečný et al., 2016; McMahan et al., 2017) keeps training data decentralized in multiple clients which collaboratively train a model under the orchestration of a server (Kairouz et al., 2021). For the server, such clients are often more unpredictable and especially more vulnerable to the attacks. Secure aggregation protocols (Bonawitz et al., 2017; So et al., 2020) ensure that the server computes aggregated updates without revealing the original data. In this paper, we focus on training-time attacks and corresponding aggregation rules.

**Training-time attacks.** Standard attacks can be broadly classified into training-time attacks (poisoning attacks) (Huang et al., 2011; Biggio et al., 2012; Mei & Zhu, 2015; Li et al., 2016; Alfeld et al., 2016; Koh & Liang, 2017; Jagielski et al., 2018; Bhagoji et al., 2019; Mahloujifar et al., 2019; Gu et al., 2019; Xie et al., 2019a; Wang et al., 2020; Allen-Zhu et al., 2021; Yang & Li, 2021; Karimireddy et al., 2021; Data & Diggavi, 2021; Carlini & Terzis, 2022) and inference-time attacks (evasion attacks) (Goodfellow et al., 2014; Carlini & Wagner, 2017). Because the server in FL trains the model across various unreliable clients with private datasets, FL usually suffers from training-time attacks (Biggio et al., 2012; Bhagoji et al., 2019; Sun et al., 2019; Bagdasaryan et al., 2020). A strong adversary can potentially participate in every training round, and meanwhile it can adapt its attacks to an updated model. One class of training-time attacks concerned in this work is *model update poisoning*. In model poisoning attack, an adversary can control some clients and can directly manipulate their outputs trying to bias the global model towards the opposite direction (Kairouz et al., 2021). If Byzantine clients have access to the updates of honest clients, they can tailor their attacks and make them difficult to detect (Lamport et al., 1982; Goodfellow et al., 2014; Blanchard et al., 2017; Fang et al., 2020; Xie et al., 2020a; Bagdasaryan et al., 2020).

**Robust aggregation and Byzantine resilience.** To improve robustness under general Byzantine clients, a number of robust aggregation schemes have been proposed, which are mainly inspired by robust statistics such as median-based aggregators (Chen et al., 2017; Yin et al., 2018), Krum (Blanchard et al., 2017), trimmed mean (Yin et al., 2018). Krum (Blanchard et al., 2017) and coordinate-wise median (Comed) (Chen et al., 2017; Yin et al., 2018) are two main aggregation rules used in this paper. Krum is a squared-distance-based aggregation rule and it aggregates the gradients that minimize the sum of squared distances to its $n - f - 2$ closest vectors where $n$ denotes the total number of clients and $f$ is the number of adversarial ones. Comed is a median-based aggregator and it selects the gradient closest to the median of each dimension.

Except of statistical aggregation rules, there are still many related works like server-side verification, client-side self-clipping etc. From the perspective of the server, Cao & Lai (2019); Fang et al. (2020); Xie et al. (2020a); Cao et al. (2021) propose some server-side verification methods using auxiliary data. Specifically, Fang et al. (2020) assume the server has a small validation dataset and uses error rates to reject harmful gradients. In (Cao & Lai, 2019; Xie et al., 2020a), the server asks a small clean dataset from clients and filters out unreliable gradients. Cao et al. (2021) utilize the ReLU-clipped cosine-similarity between local gradients and the standard one calculated by a small clean dataset as the weight for aggregation. Moreover, Karimireddy et al. (2021) and Alistarh et al. (2018) propose history-aided aggregators, and an expandable framework proposed by Ramezani-Kebrya et al. (2022) utilizes randomization to improve robustness. None of them selects a proper aggregation rule proactively during training as our framework RobustTailor. We note that all aggregation rules shown here can be added to the pool of RobustTailor because a public dataset is available in our assumption and any aggregation rule can use it. In addition, client-side clipping methods are proposed by Sun et al. (2021) and Sun et al. (2019), and client-side momentum SGD is considered by Karimireddy et al. (2021) and El Mhamdi et al. (2021). However, the ability of clients is not the focus of our paper and we will consider it in future work.

Past work has shown that these aggregators can defend successfully under specific conditions (Su & Vaidya, 2016; Blanchard et al., 2017; Chen et al., 2017). However, Fang et al. (2020) and Xie et al. (2020a) argue

that Byzantine-resilient aggregators can fail when an informed adversary tailors a careful attack. Therefore, developing a robust and efficient algorithm under such strong tailored attacks is essential to improve security of FL, which is the goal of this paper.

**Public dataset in FL.** Section 3.1.1 of (Kairouz et al., 2021) points out that a small global shared dataset is acceptable in FL. This dataset may originate from a publicly available proxy data source, a separate dataset from the clients' data that is not privacy sensitive, or perhaps a distillation of the raw data following (Wang et al., 2018). And the server could process or choose the public dataset to achieve privacy protection, e.g. producing a privacy-preserving dataset before data donation (El Ouadrhiri & Abdelhadi, 2022) and using out-of-domain data as a clean training dataset of the server (Jia et al., 2022). Note that the existence of such public dataset is a valid and common assumption in FL (Yoshida et al., 2020; Kairouz et al., 2021; Fang & Ye, 2022; Huang et al., 2022), and Sageflow (Park et al., 2021), Zeno (Xie et al., 2019b), and Zeno++ (Xie et al., 2020b) also utilize public data at the server to handle adversaries.

In addition, the presence of public data at the server makes it possible that multiple clients collaboratively train a model with formal differential or hybrid differential privacy guarantees. One direction to improve the utility of differentially private machine learning is using public data that is not subject to any privacy constraint (Gu et al., 2023). Moreover, Section 6.3 of (Kairouz et al., 2021) states that "hybrid differential privacy" is a strategy for differentially-private learning in FL, in which some users donate data with lesser privacy guarantees. Avent et al. (2017) introduce hybrid differential privacy where some opt-in users voluntarily donate data. Many companies (*e.g.,* Mozilla and Google) rely on a group of testers with higher levels of mutual trust who voluntarily opt-in to a less privacy-preserving model than that of an average end-user.

**Game theory in robust FL.** Game theory is widely used in FL such as game-based incentive model for efficient FL (Kang et al., 2019; Donahue & Kleinberg, 2021), a Stackelberg game model for the transmission strategy in FL system (Feng et al., 2019), and an evolutionary game model for training strategies of the mobile devices (Zou et al., 2019). Although these work used the game theory, none of them focused on robust training in FL. Tahanian et al. (2021) propose a game-based aggregation algorithm, named GFA, to detect and discard bad updates provided by the clients. This work does not consider the tailored attacks and it constructs a mixed-strategy game between the server and each client. Unlike our framework building a game between a server and an adversary, GFA ignores the collaboration among compromised clients, which could enhance attack effect (Fang et al., 2020).

## B    Robustness of RobustTailor

In this section, we define a general robustness definition of an aggregation rule against an attack. Note that our definition covers a broad range of settings with *general pure and mixed aggregation* along with *general pure and mixed attack* strategies. Our robustness notion leads to almost sure convergence guarantees to a local minimum of $F$ in FL, which is equivalent to being immune to training-time attacks.

**Definition 4** (Robustness of an aggregator to an attack program)**.** *Let* $\mathbf{x} \in \mathbb{R}^d$ *denote a machine learning model. Let* $\mathbf{g}_i(\mathbf{x}) = \nabla F_i(\mathbf{x}) \in \mathbb{R}^d$ *be independent honest updates for* $i \in [n]$*. Let* $G(\mathbf{x})$ *denote a function that draws an honest client* $i$ *uniformly at random followed by outputting an unbiased stochastic gradient of* $\nabla F_i(\mathbf{x})$ *over that client such that* $\mathbb{E}[G(\mathbf{x})] = \nabla F(\mathbf{x})$ *where* $\mathbb{E}$ *is over both random client and samples. Let* AG *denote an arbitrary aggregation rule, which can be a mixed aggregation strategy selecting an aggregator from* $\mathcal{A} = \{\mathrm{AG}_1, \dots, \mathrm{AG}_M\}$ *based on simulation. The output of* AG *is given by* $\breve{g}(\mathbf{x}) = \mathrm{AG}(\{\mathbf{g}'\}_{i=1}^n)$*. Note that* $\{\mathbf{g}'\}_{i=1}^n$ *includes both honest and compromised updates. The compromised updates are the output of an attack program* $\mathrm{AT}(\{\mathbf{g}_i\}_{i=1}^{n-f}, \mathcal{A})$*. Note that* AT *can be a pure or mixed attack strategy.*

*The mixed aggregation rule* AG *is Byzantine-resilient to* AT *if* $\breve{g}(\mathbf{x})$ *satisfies* $\mathbb{E}[\breve{g}(\mathbf{x})]^\top \nabla F(\mathbf{x}) > 0$ *and* $\mathbb{E}[||\breve{g}(\mathbf{x})||^r] \leq K_r \mathbb{E}[||G(\mathbf{x})||^r]$ *for* $r = 2, 3, 4$ *and some constant* $K_r$*.*

Suppose $\{\eta_t\}_{t=1}^{\infty}$ in Algorithm 2 satisfies $\sum_t \eta_t = \infty$ and $\sum_t \eta_t^2 < \infty$. For a *nonconvex loss function,* which is three times differentiable with continuous derivatives, bounded from below, and satisfies global confinement assumption in (Bottou, 1998, Section 5.1), *general pure and mixed aggregation and attack strategies* satisfying Definition 4, and *general non-iid data distribution across clients*, we can establish almost

sure convergence ($\nabla F(\mathbf{x}_t) \to 0$ a.s.) of the output of AG in Algorithm 2 along the lines of (Bottou, 1998; Fisk, 1965; Métivier, 1982).

Note that to achieve $\mathbb{E}[\breve{g}(\mathbf{x})]^\top \nabla F(\mathbf{x}) > 0$ shown above, it requires both the distance between $\nabla F(\mathbf{x})$ and the estimate of the honest update $\widetilde{\mathbf{g}}$ and the distance between $\widetilde{\mathbf{g}}$ and the expected output of Algorithm 1, i.e., $\mathbb{E}[\breve{g}(\mathbf{x})]$, are small. Let $\theta_1$ denote the angle between $\nabla F(\mathbf{x})$ and $\widetilde{\mathbf{g}}$, and let $\theta_2$ denote the angle between $\widetilde{\mathbf{g}}$ and $\mathbb{E}[\breve{g}(\mathbf{x})]$, given by $\arg\cos\left(\frac{\widetilde{\mathbf{g}}^\top \nabla F(\mathbf{x})}{\|\widetilde{\mathbf{g}}\| \cdot \|\nabla F(\mathbf{x})\|}\right)$ and $\arg\cos\left(\frac{\widetilde{\mathbf{g}}^\top \mathbb{E}[\breve{g}(\mathbf{x})]}{\|\widetilde{\mathbf{g}}\| \cdot \|\mathbb{E}[\breve{g}(\mathbf{x})]\|}\right)$, respectively. If $\theta_1 + \theta_2 < \pi/2$, then we have $\mathbb{E}[\breve{g}(\mathbf{x})]^\top \nabla F(\mathbf{x}) > 0$. Following the arguments in Appendix B, almost sure convergence of Algorithm 2 is guaranteed as long as $\theta_1 + \theta_2 < \pi/2$. This condition can be satisfied assuming 1) the public data donated by clients is representative of the underlying data distribution of honest clients, which controls $\theta_1$, and 2) the number of Byzantine clients is sufficiently small, which controls $\theta_2$. We defer derivation of the explicit necessary condition for almost sure convergence to future work.

## C Hypothetical process of model aggregation

The hypothetical process of model aggregation with $T$ rounds is shown in Algorithm 3.

---
**Algorithm 3** Hypothetical process of aggregation
---
**Input:** Initial weight vector $\mathbf{x}_0$, learning rate $\eta_t$, iteration rounds $T$, number of clients $n$, set of aggregation rules $\mathcal{A}$, set of attack algorithms $\mathcal{F}$.
**for** $t = 1$ **to** $T$ **do**
    Server sends $\mathbf{x}_t$ to all clients.
    **for** $i = 1$ **to** $n - f$ **do**
        | Honest client $i$ computes local gradient $\mathbf{g}_i(\mathbf{x}_t)$.
    Compromised clients send attacks $\mathrm{AT}^t(\{\mathbf{g}_i\}_{i=1}^{n-f}, \mathcal{A})$.
    Sever receives gradients from all clients $\{\mathbf{g}_i'\}_{i=1}^n$.
    Server chooses $\mathrm{AG}^t$ by solving $\min_{\mathrm{AG}^t \in \mathcal{A}} \max_{\mathrm{AT}^t \in \mathcal{F}} \ell(\mathrm{AG}^t, \mathrm{AT}^t, \{\mathbf{g}_i'\}_{i=1}^n)$.
    Server updates the model $\mathbf{x}_{t+1} = \mathbf{x}_t - \eta_t \mathrm{AG}^t(\{\mathbf{g}_i'\}_{i=1}^n, \mathcal{F})$.
---

## D Details of Exp3

The bandit feedback model considers the following iterate game.

**Definition 5** (Bandit setting). *The player is given a decision set $[N]$. At each iteration $k = 1, \ldots, K$:*

    *1. the player picks $i_k \in [N]$.*

    *2. the adversary picks a loss vector $\ell^k$.*

    *3. the player observes and suffers the loss at index $i_k$, i.e. $\ell^k(i_k)$.*

Exp3, as shown abstractly in Algorithm 4, enjoys a so called no-regret property in this setting. We employ Exp3 from both the perspective of a simulated server and simulated attacker to find a robust aggregation rule in Algorithm 1. In Appendix F we show how to convert the no-regret properties into a convergence guarantee.

## E Simulation of Adversary

In this section, we show the simulation of the adversary. We term *adversarial simulation* as AttackTailor, which outputs an appropriate AT at each iteration. The specific steps from the perspective of the adversary is shown in Algorithm 5. After observing $n - f$ honest gradients, the server performs $K$-round simulation and obtains a final probability distribution $\mathbf{q}$. By sampling from $\mathbf{q}$, the server selects an attack. Then, $f$ Byzantine clients create and send the compromised gradients to the server. The steps for simulating the attack procedure are summarized in Algorithm 6.

---

**Algorithm 4** Exp3

---

**Input:** Updating rate $\lambda$ and $\tilde{\lambda}$, iteration rounds $K$, $N$
Initialize weight vector $w^0(i) = 1$ for $i = 1, \ldots, N$.
**for** $k = 1$ **to** $K$ **do**

Set $W^k = \sum_{i=1}^{N} w^k(i)$, and set for $i = 1, \ldots, N$

$$\mathbf{p}(i) = (1 - \lambda)\frac{w^k(i)}{W^k} + \lambda\frac{1}{N}$$

Draw $i_k$ randomly according to the probabilities $\mathbf{p}$.
Receive loss $\ell^k$.
Set for $i = 1, \ldots, N$

$$\hat{\ell}^k(i) = \begin{cases} \ell^k/\mathbf{p}(i), & \text{if } i = i_k; \\ 0, & \text{otherwise.} \end{cases}$$

$$w^{k+1}(i) = w^k(i)\exp(-\tilde{\lambda}\hat{\ell}^k(i)/N).$$

---

Importantly, the main difference between the adversary's simulation compared with that of server is that the adversary does simulation based on realistic honest gradients while the server has access only to *noisy estimates* of true gradients. Hence, unlike typical games and simulation setups, the adversary has an additional advantage over the server, which is due to *information asymmetry*.

---

**Algorithm 5** AttackTailor

---

**Input:** Updating rates $\lambda_1$, $\lambda_2$, $\tilde{\lambda}_1$ and $\tilde{\lambda}_2$, simulation rounds $K$, gradients of honest clients $\{\mathbf{g}_i\}_{i=1}^{n-f}$, $\mathcal{A}$ and $\mathcal{F}$
Initialize weight vector $w_1^0(i) = 1$ for $i \in [M]$ and $w_2^0(j) = 1$ for $j \in [S]$.
**for** $k = 1$ **to** $K$ **do**

Set $\mathbf{p}^k(\widetilde{\text{AG}_i}) = (1 - \lambda_1)\frac{w^k(i)}{\sum_{i=1}^{M} w^k(i)} + \lambda_1\frac{1}{M}$ for $i \in [M]$.
Set $\mathbf{q}^k(\widetilde{\text{AT}_j}) = (1 - \lambda_2)\frac{v^k(j)}{\sum_{j=1}^{S} v^k(j)} + \lambda_2\frac{1}{S}$ for $j \in [S]$.
Sample $\text{AG}^k \sim \mathbf{p}^k$ and $\text{AT}^k \sim \mathbf{q}^k$ respectively.
Estimate the loss $\ell^k = \tilde{\ell}(\text{AG}^k, \text{AT}^k, \{\mathbf{g}_i\}_{i=1}^{n-f})$.
Set for $i = 1, \ldots, M$

$$\hat{\ell}_1^k(i) = \frac{\mathbb{I}\{\widetilde{\text{AG}_i} = \text{AG}^k\}}{\mathbf{p}^k(\widetilde{\text{AG}_i})}\ell^k, \quad w_1^{k+1}(i) = w_1^k(i)\exp(-\tilde{\lambda}_1\hat{\ell}_1^k(i)/M).$$

Set for $j = 1, \ldots, S$

$$\hat{\ell}_2^k(j) = \frac{\mathbb{I}\{\widetilde{\text{AT}_j} = \text{AT}^k\}}{\mathbf{q}^k(\widetilde{\text{AT}_j})}\ell^k, \quad w_2^{k+1}(j) = w_2^k(j)\exp(\tilde{\lambda}_2\hat{\ell}_2^k(j)/S).$$

Set for $j = 1, \ldots, S$

$$\mathbf{q}_j = \frac{\sum_{k=1}^{K}\mathbf{q}^k(\text{AT}_j)}{K}.$$

Sample $\text{AT} \sim \mathbf{q}$.
**Output:** AT.

---

---

**Algorithm 6** Adversary's attack

---

**Input:** Learning rate $\eta_t$, $n$ workers, $f$ compromised workers, iteration rounds $T$, $\mathcal{A}$ and $\mathcal{F}$

**for** $t = 1$ **to** $T$ **do**

> Observe all gradients of honest workers $\{\mathbf{g}_i\}_{i=1}^{n-f}$.
> Call Algorithm to attack $\text{AT}^t = \text{AttackTailor}(\{\mathbf{g}_i\}_{i=1}^{n-f}, \mathcal{A}, \mathcal{F})$.
> Produce $f$ gradients for compromised clients. Set for $j \in [f]$
>
> $$\mathbf{b}_j = \text{AT}^t(\{\mathbf{g}_i\}_{i=1}^{n-f}, \mathcal{A}).$$
>
> Send compromised gradients $\{\mathbf{b}_j\}_{j=1}^f$ to the server.

---

## F  Lemma 2 and Lemma 3

The argument builds on the well-known idea of performing approximate equilibrium computations through simultaneously played online algorithms. Let us consider a general objective, $L : [M] \times [S] \to \mathbb{R}_+$. Consider simultaneously running two algorithms on the objective $L$, such that their respective expected regrets are upper bounded by some quantities $\mathcal{R}_K^i$ and $\mathcal{R}_K^j$, i.e.,

$$\mathbb{E}\left[\sum_{k=1}^K L(i_k, j_k) - \sum_{k=1}^K L(i, j_k)\right] \le \mathcal{R}_K^i, \quad \mathbb{E}\left[\sum_{k=1}^K L(i_k, j) - \sum_{k=1}^K L(i_k, j_k)\right] \le \mathcal{R}_K^j, \tag{10}$$

for any $i \in [M]$ and $j \in [S]$ where the expectation is taken over the randomness of the algorithms.

**Lemma 2** (Folklore)**.** *Suppose we run two algorithms simultaneously with regrets as in* (10) *to obtain* $\{(i_k, j_k)\}_{k=1}^K$. *By playing $\bar{i}$ uniformly sampled from $\{i_k\}_{k=1}^K$, we guarantee that*

$$\mathbb{E}_{\bar{i}}\left[L(\bar{i}, j)\right] \le \mathbb{E}_{i^\star \sim \mathbf{p}^\star, j^\star \sim \mathbf{q}^\star}\left[L(i^\star, j^\star)\right] + \frac{1}{K}(\mathcal{R}_K^i + \mathcal{R}_K^j), \tag{11}$$

*for any $j \in [S]$ where $(\mathbf{p}^\star, \mathbf{q}^\star)$ is a Nash equilibrium of $\mathbb{E}_{i^\star \sim \mathbf{p}^\star, j^\star \sim \mathbf{q}^\star}\left[L(i^\star, j^\star)\right]$.*

This kind of result is well-known in the literature (see for instance Dughmi et al. (2017, Cor. 4) for a very related result).

*Proof.* Defined $\bar{i} \sim \bar{\mathbf{p}}$ to be uniformly sampled from $\{i_k\}_{k=1}^K$, and $\bar{j} \sim \bar{\mathbf{q}}$ to be uniformly sampled from $\{j_k\}_{k=1}^K$. Using the no-regret property from (10),

$$\begin{aligned}
\mathbb{E}[L(\bar{i}, j)] = \mathbb{E}\left[\frac{1}{K}\sum_{k=1}^K L(i_k, j)\right] &\le \mathbb{E}\left[\frac{1}{K}\sum_{k=1}^K L(i_k, j_k)\right] + \frac{1}{K}\mathcal{R}_K^j \\
\mathbb{E}[L(i, \bar{j})] = \mathbb{E}\left[\frac{1}{K}\sum_{k=1}^K L(i, j_k)\right] &\ge \mathbb{E}\left[\frac{1}{K}\sum_{k=1}^K L(i_k, j_k)\right] - \frac{1}{K}\mathcal{R}_K^i,
\end{aligned} \tag{12}$$

for any $i \in [M]$ and $j \in [S]$, where the expectation is taken over $\bar{i}$ and $\bar{j}$ and the randomness of the algorithms. Subtracting the two equations,

$$\mathbb{E}[L(\bar{i}, j)] - \mathbb{E}[L(i, \bar{j})] \le \frac{1}{K}(\mathcal{R}_K^j + \mathcal{R}_K^i) =: \varepsilon_{\text{sim}}. \tag{13}$$

Observe that by first evoking the inequality with $i \sim \bar{\mathbf{p}}$ and secondly with $j \sim \bar{\mathbf{q}}$, we see that $(\bar{\mathbf{p}}, \bar{\mathbf{q}})$ is an $\varepsilon_{\text{sim}}$-approximate Nash equilibrium, i.e.,

$$\mathbb{E}[L(\bar{i}, j)] - \varepsilon_{\text{sim}} \le \mathbb{E}[L(\bar{i}, \bar{j})] \le \mathbb{E}[L(i, \bar{j})] + \varepsilon_{\text{sim}}. \tag{14}$$

We are interested in the $i$-players performance $\mathbb{E}[L(\bar{i}, j)]$ which we can relate to the mixed strategy Nash equilibrium defined as $\mathbb{E}[L(i^\star, j)] \leq \mathbb{E}[L(i^\star, j^\star)] \leq \mathbb{E}[L(i, j^\star)]$ where $i^\star \sim \mathbf{p}^\star$ and $j^\star \sim \mathbf{q}^\star$. By picking $i \sim \mathbf{p}^\star$ in (13) we get,

$$\begin{aligned}
\mathbb{E}[L(\bar{i}, j)] &\leq \mathbb{E}[L(i, \bar{j})] + \frac{1}{K}(\mathcal{R}_K^j + \mathcal{R}_K^i) \\
&= \mathbb{E}[L(i^\star, \bar{j})] + \frac{1}{K}(\mathcal{R}_K^j + \mathcal{R}_K^i) \\
&\leq \mathbb{E}[L(i^\star, j^\star)] + \frac{1}{K}(\mathcal{R}_K^j + \mathcal{R}_K^i),
\end{aligned}$$

(15)

where the last inequalities follows by the definition of a Nash equilibrium above. The claim follows by writing the expectation on the RHS in terms of $\mathbf{p}^\star$ and $\mathbf{q}^\star$. □

When the algorithms have sublinear regrets, we refer to them as *no-regret* algorithms. This condition ensures that the error term in (11) vanishes as $K \to \infty$. Exp3 of Auer et al. (2002), employed by both the adversary and server in Algorithm 1, enjoys such a no-regret property.

**Lemma 3** (Hazan 2016, Lemma 6.3). *Let $K$ be the horizon, $N$ be the number of actions, and $L_k : [N] \to \mathbb{R}_+$ be non-negative losses for all $k$. Then Exp3 with stepsize $\lambda = \sqrt{\frac{\log N}{KN}}$ enjoys the following regret bound,*

$$\mathbb{E}\left[\sum_{k=1}^{K} L_k(i_k) - \sum_{k=1}^{K} L_k(i)\right] \leq 2\sqrt{KN \log N},$$

(16)

*for any $i \in [N]$, where the expectation is taken over the randomness of the algorithm.*

## G   Proof of Lemma 1

*Proof.* Let both player $i$ and player $j$ in Lemma 2 employ the no-regret algorithm Exp3 such that Lemma 3 applies and consequently $\mathcal{R}_K^i$ and $\mathcal{R}_K^j$ in (10) reduce to

$$\mathcal{R}_K^i = 2\sqrt{KM \log M}, \quad \mathcal{R}_K^j = 2\sqrt{KS \log S}.$$

(17)

Substituting (17) into Lemma 2, we have

$$\mathbb{E}_{\bar{i}}\left[L(\bar{i}, j)\right] \leq \mathbb{E}_{i^\star \sim \mathbf{p}^\star, j^\star \sim \mathbf{q}^\star}\left[L(i^\star, j^\star)\right] + 2\frac{\sqrt{M \log M} + \sqrt{S \log S}}{\sqrt{K}}.$$

(18)

Notice that Algorithm 1 is an instance of two simultaneously played Exp3 algorithms where $i = \text{AG}, j = \text{AT}$ and $L(i, j) = \widetilde{\ell}\left(\text{AG}, \text{AT}, \{\widetilde{\mathbf{g}}_i\}_{i=1}^{n-f}\right)$. It follows from (18) that

$$\mathbb{E}_{\text{AG}}\left[\widetilde{\ell}\left(\text{AG}, \text{AT}, \{\widetilde{\mathbf{g}}_i\}_{i=1}^{n-f}\right)\right] \leq \mathbb{E}_{\text{AG}^\star \sim \mathbf{p}^\star, \text{AT}^\star \sim \mathbf{q}^\star}\left[\widetilde{\ell}(\text{AG}^\star, \text{AT}^\star, \{\widetilde{\mathbf{g}}_i\}_{i=1}^{n-f})\right] + \varepsilon_{\text{sim}}$$

(19)

where

$$\varepsilon_{\text{sim}} = 2\frac{\sqrt{M \log M} + \sqrt{S \log S}}{\sqrt{K}}$$

(20)

where AG is the average iterate as defined in Algorithm 1. We can concisely write the Nash equilibrium on the R.H.S. of (19) in terms of the payoff matrix $\tilde{\mathbf{L}}$ from MixedNash defined componentwise as $\tilde{\mathbf{L}}(\text{AG}, \text{AT}) = \widetilde{\ell}(\text{AG}, \text{AT}, \{\widetilde{\mathbf{g}}_i\}_{i=1}^{n-f})$. This completes the proof. □

# H  Proof of Theorem 1

*Proof.* Expanding R.H.S. of Eq. (7) and referring Lemma 1, we have

$$
\begin{aligned}
\langle \mathbb{E}_{\mathrm{AG}}[\mathrm{AG}(\{\widetilde{\mathbf{g}}_i'\}_{i=1}^n)], \widetilde{\mathbf{g}}^\star \rangle &= \frac{1}{2}\left( \|\mathbb{E}_{\mathrm{AG}}[\mathrm{AG}(\{\widetilde{\mathbf{g}}_i'\}_{i=1}^n)]\|^2 + \|\widetilde{\mathbf{g}}^\star\|^2 - \mathbb{E}_{\mathrm{AG}}[\widetilde{\ell}] \right) \\
&\geq \frac{1}{2}\left( \|\mathbb{E}_{\mathrm{AG}}[\mathrm{AG}(\{\widetilde{\mathbf{g}}_i'\}_{i=1}^n)]\|^2 + \|\widetilde{\mathbf{g}}^\star\|^2 - {\mathbf{p}^\star}^\top \tilde{\mathbf{L}} \mathbf{q}^\star - \varepsilon_{\mathrm{sim}} \right)
\end{aligned}
\tag{21}
$$

where $\widetilde{\ell} = \widetilde{\ell}(\mathrm{AG}, \mathrm{AT}, \{\widetilde{\mathbf{g}}_i\}_{i=1}^{n-f})$, $\mathrm{AG} \sim \mathbf{p}$, $\mathrm{AT} \in \mathcal{F}$, and $\{\widetilde{\mathbf{g}}_i\}_{i=1}^{n-f}$ are fixed simulated gradients. We now establish a lower bound on $\langle \mathbb{E}_{\mathrm{AG}}[\mathrm{AG}(\{\widetilde{\mathbf{g}}_i'\}_{i=1}^n)], \mathbf{g}^\star \rangle$ for the ideal update $\mathbf{g}^\star$ using the lower bound in Eq. (21).

$$
\begin{aligned}
\langle \mathbb{E}_{\mathrm{AG}}[\mathrm{AG}(\{\widetilde{\mathbf{g}}_i'\}_{i=1}^n)], \mathbf{g}^\star \rangle &= \langle \mathbb{E}_{\mathrm{AG}}[\mathrm{AG}(\{\widetilde{\mathbf{g}}_i'\}_{i=1}^n)], \mathbf{g}^\star - \widetilde{\mathbf{g}}^\star \rangle + \langle \mathbb{E}_{\mathrm{AG}}[\mathrm{AG}(\{\widetilde{\mathbf{g}}_i'\}_{i=1}^n)], \widetilde{\mathbf{g}}^\star \rangle \\
&\geq \langle \mathbb{E}_{\mathrm{AG}}[\mathrm{AG}(\{\widetilde{\mathbf{g}}_i'\}_{i=1}^n)], \widetilde{\mathbf{g}}^\star \rangle - \|\mathbb{E}_{\mathrm{AG}}[\mathrm{AG}(\{\widetilde{\mathbf{g}}_i'\}_{i=1}^n)]\| \|\mathbf{g}^\star - \widetilde{\mathbf{g}}^\star\| \\
&\geq \frac{1}{2}\big( \|\mathbb{E}_{\mathrm{AG}}[\mathrm{AG}(\{\widetilde{\mathbf{g}}_i'\}_{i=1}^n)]\|^2 - {\mathbf{p}^\star}^\top \tilde{\mathbf{L}} \mathbf{q}^\star - \varepsilon_{\mathrm{sim}} \\
&\quad + \|\widetilde{\mathbf{g}}^\star\|^2 - 2\|\mathbb{E}_{\mathrm{AG}}[\mathrm{AG}(\{\widetilde{\mathbf{g}}_i'\}_{i=1}^n)]\| \|\mathbf{g}^\star - \widetilde{\mathbf{g}}^\star\| \big) \\
&\geq \frac{1}{2}\left( \|\widetilde{\mathbf{g}}^\star\|^2 - C\|\widetilde{\mathbf{g}}^\star - \mathbf{g}^\star\|^2 - {\mathbf{p}^\star}^\top \tilde{\mathbf{L}} \mathbf{q}^\star - \varepsilon_{\mathrm{sim}} \right)
\end{aligned}
\tag{22}
$$

where $C$ is an upper bound on the norm of the output of AG.

Under Assumption 4, we have

$$
\mathbb{E}\|\widetilde{\mathbf{g}}^\star - \mathbf{g}^\star\|^2 \leq \frac{\mathsf{V}_{\mathrm{est}}}{n-f} + \mathsf{B}_{\mathrm{est}}.
\tag{23}
$$

Then, take expectation from both sides of Eq. (22):

$$
\langle \mathbb{E}_{\mathrm{AG}}[\mathrm{AG}(\{\widetilde{\mathbf{g}}_i'\}_{i=1}^n)], \mathbf{g}^\star \rangle \geq \frac{1}{2}\left( \mathbb{E}\|\widetilde{\mathbf{g}}^\star\|^2 - \frac{C\mathsf{V}_{\mathrm{est}}}{n-f} - \mathbb{E}[{\mathbf{p}^\star}^\top \tilde{\mathbf{L}} \mathbf{q}^\star] - 2\frac{\sqrt{M\log M} + \sqrt{S\log S}}{\sqrt{K}} - C\mathsf{B}_{\mathrm{est}} \right).
\tag{24}
$$

If $\langle \mathbb{E}_{\mathrm{AG}}[\mathrm{AG}(\{\widetilde{\mathbf{g}}_i'\}_{i=1}^n)], \mathbf{g}^\star \rangle \geq 0$, then the almost sure convergence $(\nabla F(\mathbf{x}_t) \to 0 \text{ a.s.})$ of the output of AG in Algorithm 2 can be established along the lines of (Bottou, 1998; Fisk, 1965; Métivier, 1982), which complete the proof of Theorem 1. $\qquad\square$

# I  Privacy Guarantee

To further settle privacy concerns, we also consider the second scenario under stringent privacy requirements. The server trusts only a small subset of honest clients, which can provide reliable updates. It is normal that the server has some core clients that are trustworthy. For example, some companies rely on testers with high-level mutual trust. In such a situation, RobustTailor does not lead to any additional privacy loss compared to any other aggregation method that receives individual stochastic gradients from clients. Note that RobustTailor is compatible with *all privacy-preserving techniques, e.g.,* differential privacy (DP) (Bassily et al., 2014; Wei et al., 2020), homomorphic encryption (Aono et al., 2017), and secure multiparty computation (Mohassel & Zhang, 2017). Let $r_{\max}$ denote the largest sampling ratio among trusted clients. Along the lines of, e.g., [Abadi et al., 2016b, Theorem 1], clipping and adding zero-mean Gaussian noise with a standard deviation $\Omega(r_{\max}\sqrt{T\log(1/\delta)}\log(T/\delta)/\epsilon)$ to updates of trusted clients is *sufficient to guarantee* $(\epsilon, \delta)$-DP. Similar noise can be added to all clients to extend privacy guarantees to the entire group, as discussed in Corollary 1.

**Corollary 1** (Joint convergence and privacy guarantees)**.** *Under the setting described in Theorem 1, clipping and adding zero-mean Gaussian noise with a standard deviation $\Omega(r_{\max}\sqrt{T\log(1/\delta)}\log(T/\delta)/\epsilon)$, and the first term of R.H.S of Eq. (9) dominates the lower bound, then $n - f = \Omega(r_{\max}^2 T\log(1/\delta)\log(T/\delta)/\epsilon^2)$ is sufficient for RobustTailor to successfully establish both almost sure convergence and $(\epsilon, \delta)$-DP guarantees.*

**Remark 2** (Trade-off between privacy and convergence guarantees)**.** *There exists a trade-off between privacy and convergence guarantees. In particular, by decreasing $\epsilon, \delta$ in $(\epsilon, \delta)$-DP, stronger privacy guarantees are achieved for trusted clients; while it slows down convergence in Theorem 1 due to an increased $\mathsf{V}_{\mathrm{est}}$.*

## J Experimental Details and Additional Experiments

In this section, we provide the training hype-parameters and show a series of additional experiments.

### J.1 Details of Implementation

Both MNIST (Lecun et al., 1998) and FMNIST (Xiao et al., 2017) datasets contain 60000 training samples and 10000 test samples. Each sample is a 28 by 28 pixel grayscale image. The details of training hype-parameters are shown in Table 1. The network architecture is a fully connected neural network with two fully connected layers (Leroux et al., 2016). The number of neurons is 100 and 10 for the first and second layer, respectively. All experiments have been run on a cluster with Xeon-Gold processors and V100 GPUs.

Table 1: Training hyper-parameters for MNIST, FMNIST, and CIFAR10

| Hyper-parameter | MNIST | FMNIST | CIFAR10 |
|---|---|---|---|
| Learning Rate | 0.01 | 0.003 | 0.002 |
| Batch Size | 50 | 50 | 80 |
| Total Iterations | 15K | 10K | 10K |
| $K$ | 10 | 10 | 10 |
| $\lambda_1, \lambda_2$ | 0.3 | 0.3 | 0.3 |
| $\tilde{\lambda}_1, \tilde{\lambda}_2$ | 0.3 | 0.3 | 0.3 |

### J.2 Additional Experiments

In this section, we set up two additional experiments to further validate the performance of RobustTailor.

**Different datasets.** We train a CNN model on MNIST (Lecun et al., 1998), Fashion-MNIST (FMNIST) (Xiao et al., 2017) and CIFAR10 (Krizhevsky & Hinton, 2009) under iid setting. We summarize the training results against 3 attacks in Fig. 10 and they are consensus with the results shown in Section 6.

**More Byzantines.** Fig. 11 shows the results when there are 4 Byzantines (2 Byzantines in the basic setting) in 12 total clients under three different attacks. We observe that both Krum and Comed are sensitive to the number of Byzantine clients while RobustTailor is much more stable. Specifically, Krum has lower accuracy closing to zero, and Comed shows more obvious fluctuations.

**Non-iid settings.** We also extend our consideration to more realistic settings with non-iid data distribution across clients. We use the *heterogeneous degree* $\mu \in [0, 1]$ to represent the level of disparity among clients' local data. To be specific, we construct a setting, in which $100\mu\%$ of local data for each client is drawn in a non-identical but independent manner from a particular class corresponding to the client index and $100(1-\mu)\%$ of the local data is drawn iid from all classes. A small $\mu$ represents low disparity while a large $\mu$ means significant disparity among clients. Fig. 12 shows three non-iid settings including $\mu = 0.1, 0.5, 0.9$. RobustTailor, with the most basic pool of robust aggregators that are not designed to address non-iid settings, shows a satisfactory level of robustness even under heterogeneous data settings.

**Combination strategies of aggregators.** To further validate the effectiveness of RobustTailor, we compare it with other three strategies combining aggregators: 1) MixTailor (Ramezani-Kebrya et al., 2022): randomly choose an aggregator; 2) Average: average of the aggregation results from multiple aggregation rules; 3) Closest: choose an aggregation result that is closest to $\tilde{g}^\star$ from multiple aggregation rules. Fig. 13 demonstrate that RobustTailor always outperforms these three combinations of aggregators, which further indicates that RobustTailor can select a proper aggregator out at each iteration.

## K Computational Complexity

The computational complexity bound depends on the simulation of the inner loop (including simulation rounds $K$, aggregator set $\mathcal{A}$, and attack set $\mathcal{F}$) and problem dimensions of the outer loop (including number

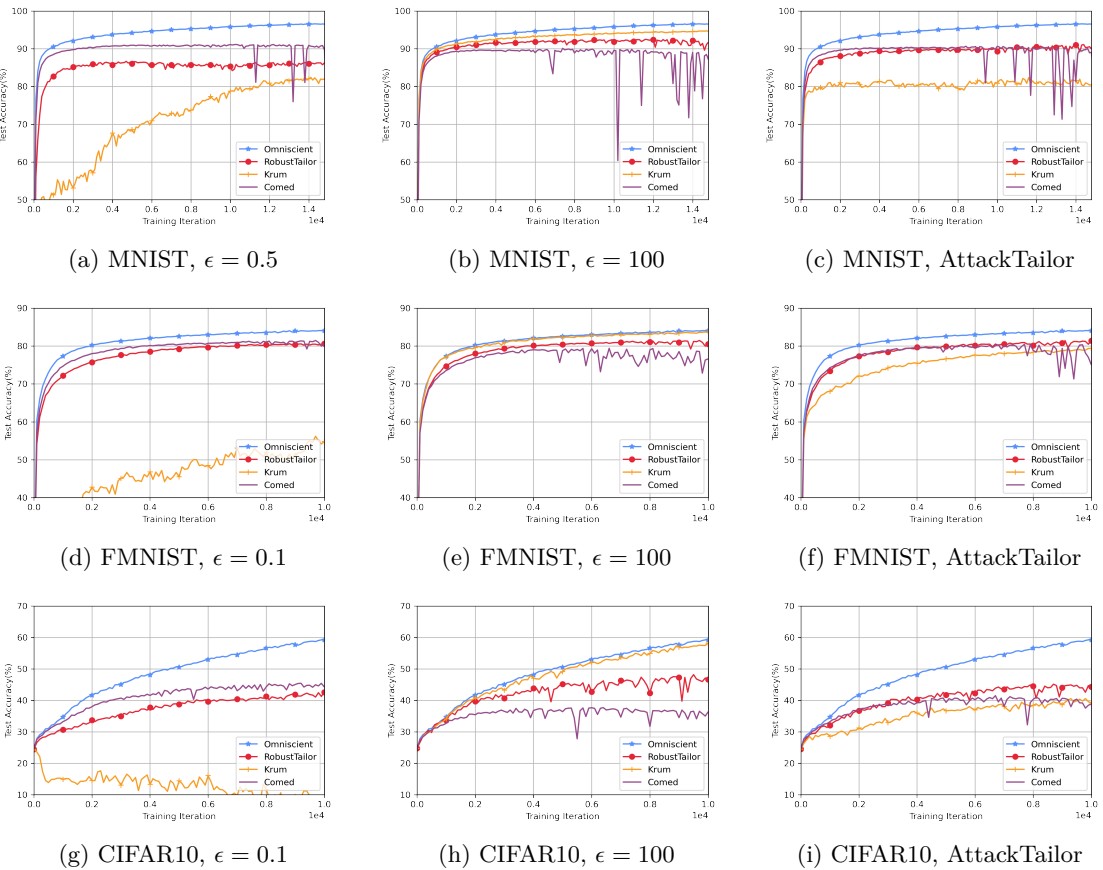

Figure 10: **iid setting on three datasets.** RobustTailor includes Krum and Comed. AttackTailor includes $\epsilon = 0.1/0.5$ and $\epsilon = 100$.

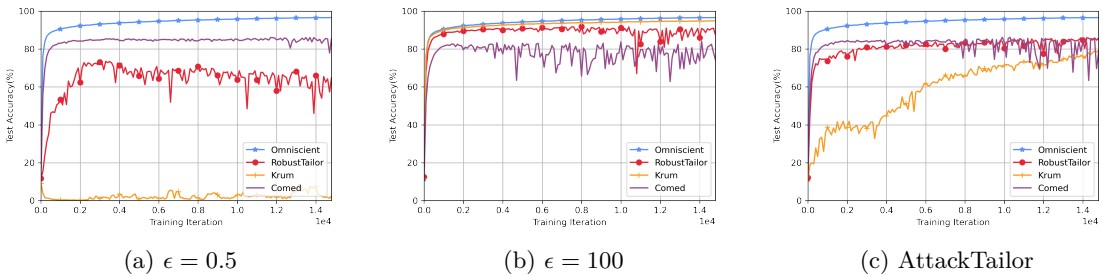

Figure 11: **4 Byzantines.** There are $n = 12$ total workers including $f = 4$ Byzantine workers.

of clients $n$ and the dimension of gradients). We show the theoretical analysis below and utilize empirical results to prove that it is worth to trade a little more computational complexity for a great robust model.

**Theoretical analysis** We first analyze the time complexity of RobustTailor per simulation round. The computational cost of RobustTailor is influenced by the server's aggregation rules and the adversary's attacks. If $n$ clients submit $d$-dimensional vectors, Krum's expected time complexity is $\mathcal{O}(n^2 d)$ (Blanchard et al., 2017) and Comed's is $\mathcal{O}(nd)$ (Pillutla et al., 2022).

For more fine-grained complexity analysis, the complexity depends linearly on $M$ and $S$, but they are not inherently sequential, so their dependency can potentially be parallelized away. Suppose $\{\tilde{T}_1, \ldots, \tilde{T}_M\}$ denote

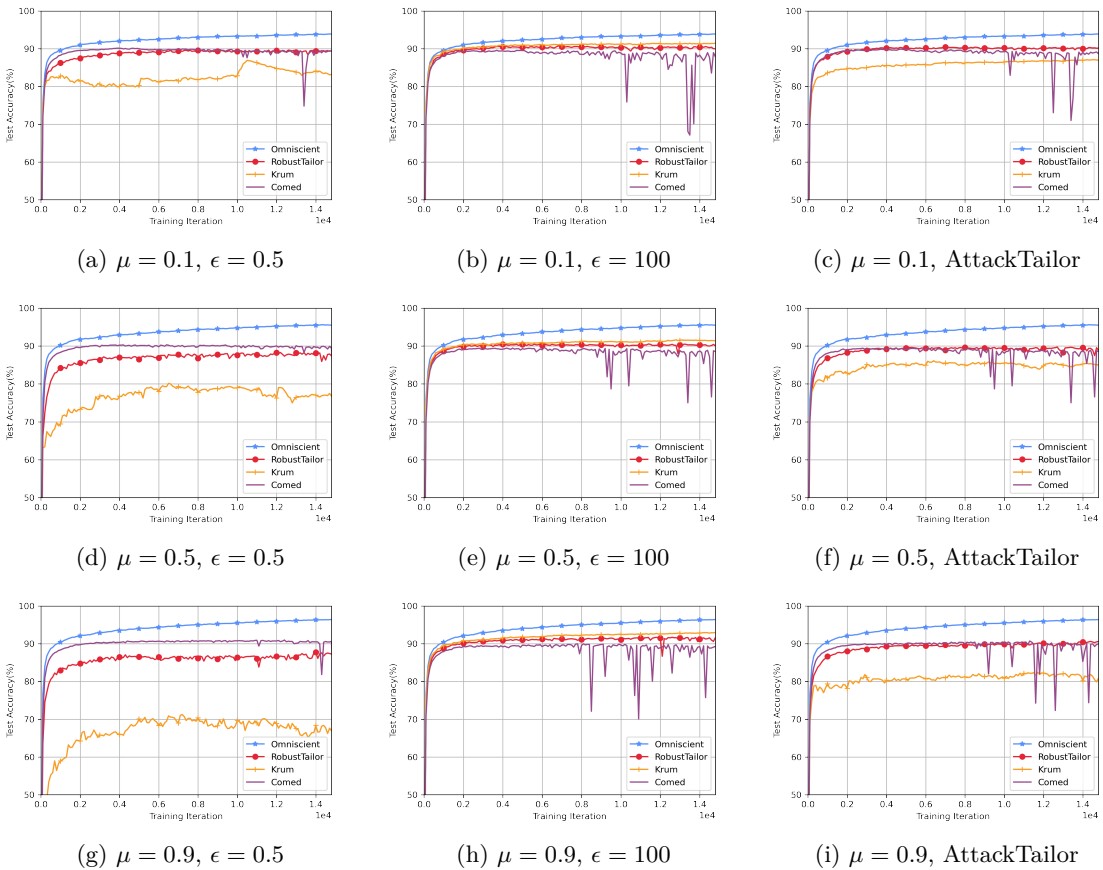

(a) $\mu = 0.1$, $\epsilon = 0.5$      (b) $\mu = 0.1$, $\epsilon = 100$      (c) $\mu = 0.1$, AttackTailor

(d) $\mu = 0.5$, $\epsilon = 0.5$      (e) $\mu = 0.5$, $\epsilon = 100$      (f) $\mu = 0.5$, AttackTailor

(g) $\mu = 0.9$, $\epsilon = 0.5$      (h) $\mu = 0.9$, $\epsilon = 100$      (i) $\mu = 0.9$, AttackTailor

Figure 12: **Non-iid setting.** Larger $\mu$ means higher heterogeneous degree.

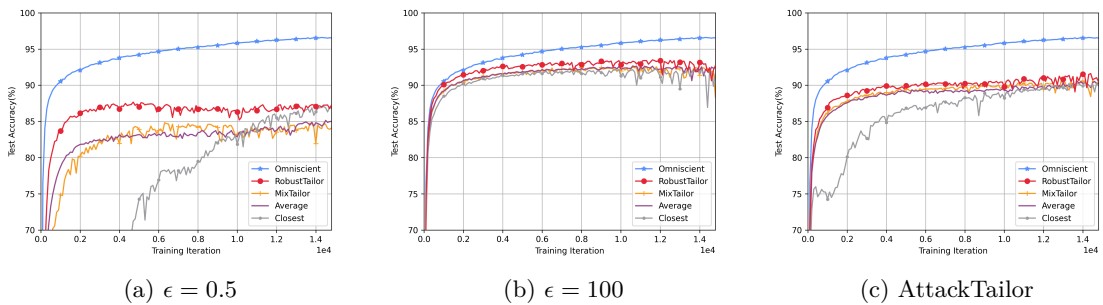

(a) $\epsilon = 0.5$      (b) $\epsilon = 100$      (c) AttackTailor

Figure 13: **Combination strategies of aggregators.**

the number of elementary operations to run each aggregation rule within the set of $M$ aggregation rules. The *worst-case* runtime complexity of RobustTailor per simulation round is determined by $\max_{i \in [M]} \tilde{T}_i$. However, the average complexity per round is the expected value of the number of elementary operations where the expectation is over the distribution of how likely each aggregator is chosen during simulation, which can be estimated empirically. Let us use $\tilde{p}_i$ to denote the probability of choosing $\mathcal{A}_i$. The average complexity per round is given by $\overline{T} = \sum_{i=1}^{M} \tilde{T}_i \tilde{p}_i$. Finally, the overall time complexity of RobustTailor with $K$ simulation rounds is given by $\mathcal{O}((M + S)\overline{T}K)$.

Moreover, the number of elementary operations for simulation can be much smaller than applying the actual aggregator on the model during training assuming the size of the public data is very small, which is typically

the case in practice(Zhao et al., 2018; Yoshida et al., 2020). Note that our algorithm just adds computation complexity to the server while all clients remain the same cost based on their models and datasets. Therefore, it is worthwhile to trade slightly longer training time for a significantly improved training procedure w.r.t. robustness.

**Empirical results** We show computation costs and accuracy for different aggregation rules after running 15k iterations in reality, whose results are also shown in Fig. 1. We can see that RobustTailor still maintains a stable and high accuracy when facing a powerful adversary although it needs more computation time. However, Krum cannot reach a high accuracy and Comed shows a very unstable performance with lots of fluctuations when facing a strong adversary with AttackTailor. We note that compared with undesirable models of Krum and Comed, RobustTailor improves accuracy and stability drastically at the cost of slightly increased training time.

Table 2: Computational complexity based on MNIST after running 15k iterations

| Aggregator | Time | Accuracy |
|---|---|---|
| Omniscient | 34 min | 96.63% |
| RobustTailor | 96 min | 85.87% |
| Krum | 37 min | 82.13% |
| Comed | 52 min | 90.74% |

(a) $\epsilon = 0.5$

| Aggregator | Time | Accuracy |
|---|---|---|
| Omniscient | 34 min | 96.63% |
| RobustTailor | 96 min | 92.03% |
| Krum | 37 min | 94.74% |
| Comed | 52 min | 60.39%-88.80% |

(b) $\epsilon = 100$

| Aggregator | Time | Accuracy |
|---|---|---|
| Omniscient | 34 min | 96.63% |
| RobustTailor | 190 min | 89.72% |
| Krum | 90 min | 80.98% |
| Comed | 121 min | 71.34%-89.75% |

(c) AttackTailor

