# OpenReview forum: "Mixed Nash for Robust Federated Learning"
_TMLR — Accepted by TMLR_

### Review · Reviewer_BVaL · 2023-12-03

**Summary Of Contributions:**

This paper tackles the Byzantine-robustness problem in federated learning. Instead of designing one robust aggregation rule that is robust to all attackers, this paper studies how to "ensemble" multiple imperfect robust aggregation rules, where each of them is only immune to a subset of attacks. The proposed algorithm has theoretical robustness guarantee, and is tested under extensive experiment settings.

**Audience:**

Yes

**Claims And Evidence:**

Yes

**Requested Changes:**

- Novelty of game between server and adversary (Section 3.1). Although this perspective is novel to a certain extent, here are some other works using this idea [1, 2]. If the authors insist that this work is the first of its kind, I think a detailed comparison to these related papers can be helpful to support such a claim.
- Extend the experiments in (1) hybrid attacks and (2) more baselines, according to my comments in weaknesses.

[1] Ali Ramezani-Kebrya, Iman Tabrizian, Fartash Faghri, Petar Popovski. MixTailor: Mixed Gradient Aggregation for Robust Learning Against Tailored Attacks. Trans. Mach. Learn. Res. 2022 (2022)

[2] E. Tahanian, M. Amouei, H. Fateh, M. Rezvani. A Game-theoretic Approach for Robust Federated Learning. International Journal of Engineering. 2021.



**Minor Changes**:

1. Page 1, paragraph 3, line 2: the the -> the
2. Page 3 paragraph "adversary's goal", line 3: double commas

**Strengths And Weaknesses:**

**Strengths**

1. The prospective of combining multiple "weak" robust aggregator and optimizing the strategy to choosing them is interesting and novel.
2. The paper is well-written. The description of the setting, problem formulation, and the algorithm is very clear.
3. The author provide valid convergence guarantee of the algorithm.
4. The experiments include a wide range of FL settings: partial participation, dynamic number of attackers, partial knowledge, imperfect public dataset, etc.

**Weaknesses**

1. This paper assume that the adversary will randomly draw one kind of attack from F, i.e., the pool of possible attack methods, with probability distribution q. Although the strategy is mixed, instead of deterministic, it is (implicitly) assumed that the adversary will use the same type of attack for all Byzantine clients. However in real scenarios, there could be multiple types of attacks simultaneously. If no aggregation rule is immune to all attack algorithms, it is skeptical whether a random choice of them can resist this hybrid of attack, especially given the fact that the server can only choose one aggregation rule at each round.
2. The novelty of this paper is the way to combine multiple weak aggregator. However, the author only compare it with very simple baselines: Krum and CoMed. The author should also compare the proposed method with other combination of weak aggregator. To name a few:
   2.1. MixTailor [1]: randomly choose an aggregator.
   2.2. Average of the aggregation results from multiple aggregation rules.
   2.3. Choose an aggregation result that is closer to $\tilde{g}^*$ from multiple aggregation rules.
3. Moreover, since RobustTailor uses public data, it should be compared to the baseline AGRs that use public data. For instance, Zeno [2], FLTrust [3].

**Minor Questions**

When using a clean public dataset, it is claimed in some works, e.g. [3], that n >= 2f + 1 is not necessary anymore. Is it possible to get rid of this assumption for your algorithm? This is an open question and does not change my decision to this paper.



References.

[1] Ali Ramezani-Kebrya, Iman Tabrizian, Fartash Faghri, Petar Popovski. MixTailor: Mixed Gradient Aggregation for Robust Learning Against Tailored Attacks. Trans. Mach. Learn. Res. 2022 (2022)

[2] Cong Xie, Sanmi Koyejo, Indranil Gupta. Zeno: Distributed Stochastic Gradient Descent with Suspicion-based Fault-tolerance. ICML 2019.

[3] Xiaoyu Cao, Minghong Fang, Jia Liu, Neil Zhenqiang Gong. FLTrust: Byzantine-robust Federated Learning via Trust Bootstrapping. NDSS 2021.

---

> ### Author Response · Authors · 2024-01-10
> **Response to Reviewer BVaL (Part 1)**
>
> We thank the reviewer for their valuable feedback and address all remaining concerns below:
>
> > Q1. This paper assume that the adversary will randomly draw one kind of attack from $\mathcal{F}$, i.e., the pool of possible attack methods, with probability distribution $q$. Although the strategy is mixed, instead of deterministic, it is (implicitly) assumed that the adversary will use the same type of attack for all Byzantine clients. However in real scenarios, there could be multiple types of attacks simultaneously. If no aggregation rule is immune to all attack algorithms, it is skeptical whether a random choice of them can resist this hybrid of attack, especially given the fact that the server can only choose one aggregation rule at each round.
>
> A1. The assumption of all Byzantine clients using the same attack at each iteration ensures the strength of the attack, which aligns with common assumptions in existing works [Fang et al., 2020; Ramezani-Kebrya et al., 2022; Karimireddy et al., 2022]. Additionally, previous research, exemplified by Figure 4 in [Ramezani-Kebrya et al., 2022], suggests that a hybrid attack might not be as effective. This corroborates the notion that a training-time attack should be consistently applied by all Byzantine clients in collaboration [Blanchard et al., 2017; El Mhamdi et al., 2018].
>
> We believe that $\mathrm{RobustTailor}$ can still select the most robust $\mathrm{AG}$, even though there is no universal aggregating rule that is robust across all attacks. To further verify, we define a hyrid attack that one Byzantine client uses $\epsilon=0.5$ attack and another compromised client uses $\epsilon=100$ attack. [The empirical result](https://imgur.com/a/PQjPQQT) demonstrates that $\mathrm{RobustTailor}$ continues to outperform Krum and Comed under such conditions.
>
>
>
> > Q2. The novelty of this paper is the way to combine multiple weak aggregator. However, the author only compare it with very simple baselines: Krum and CoMed. The author should also compare the proposed method with other combination of weak aggregator. To name a few: 2.1. MixTailor: randomly choose an aggregator. 2.2. Average of the aggregation results from multiple aggregation rules. 2.3. Choose an aggregation result that is closer to $\tilde{g}^\star$ from multiple aggregation rules.
>
> A2. Thanks for the reviewer's suggestion. We provide additional results [here](https://imgur.com/a/UYZMMsV) and we also add them in Appendix J.2. The figures demonstrate that $\mathrm{RobustTailor}$ always outperforms these three combinations of aggregators, which further indicates that $\mathrm{RobustTailor}$ can select a proper aggregator out at each iteration.
>
>
>
> > Q3. Moreover, since RobustTailor uses public data, it should be compared to the baseline AGRs that use public data. For instance, Zeno, FLTrust.
>
> A3. We assert that our comparison includes error rate-based rejection (ERR) and loss function-based rejection (LFR) towards the end of Section 6.4. As for comparisons with more aggregators utilizing auxiliary data, we didn't conduct direct comparisons as RobustTailor is adaptable and compatible with nearly all aggregators, including Zeno and FLTrust. Notably, existing studies have indicated vulnerabilities in Zeno and FLTrust to specific attacks. For instance, Figures 4 and 5 in [Park et al., 2021] demonstrate the potential susceptibility of Zeno to certain model poisoning attacks, while Figures 2 and 3 in [1] showcase vulnerabilities of FLTrust to Alittle and inner product manipulation attacks.
>
> Moreover, as an additional practical evaluation, we conducted tests including Zeno. [The empirical result](https://imgur.com/a/I1vT53e) demonstrate that an $\epsilon=1$ attack negatively impacts Zeno, whereas RobustTailor, inclusive of Krum, CoMed, and Zeno, assists in defense against such attacks. Therefore, we posit that RobustTailor holds the potential to augment these foundational aggregators.
>
>
> [1] Wei Wan, Shengshan Hu, Minghui Li, Jianrong Lu, Longling Zhang, Leo Yu Zhang, and Hai Jin. A four-pronged defense against Byzantine attacks in federated learning. In Proceedings of the 31st ACM International Conference on Multimedia, 2023.

---

> ### Author Response · Authors · 2024-01-10
> **Response to Reviewer BVaL (Part 2)**
>
> > Q4. When using a clean public dataset, it is claimed in some works, e.g. FLTrust, that $n \geq 2f + 1$ is not necessary anymore. Is it possible to get rid of this assumption for your algorithm? This is an open question and does not change my decision to this paper.
>
> A4. $\mathrm{RobustTailor}$ is more accurately described as a **framework** for aggregators rather than a new aggregation rule. Therefore, it depends on aggregators of the server. We set $n \geq 2f + 1$ just because many common aggregators require it such as Krum, Bulyan. Table 1 in [2] shows more information about the number of malicious clients in different aggregators.
>
>
> > Q5. Novelty of game between server and adversary (Section 3.1). Although this perspective is novel to a certain extent, here are some other works using this idea (MixTailor and GFA). If the authors insist that this work is the first of its kind, I think a detailed comparison to these related papers can be helpful to support such a claim.
>
> A5. A more precise contribution of our work is: $\mathrm{RobustTailor}$ is the first framework that selects suitable aggregators actively by framing robust learning problem under training-time tailored attacks as a game. We have revised Section 3.1 to make this clearer.
> MixTailor is also a framework but it simply picks an aggregator randomly. GFA frames a game between the server and each client and aims to distinguish good or bad clients through updates of clients, and importantly GFA is an aggregation rule rather than a framework thus it cannot choose from existing aggregators and it can be incorporated within $\mathrm{RobustTailor}$ instead.
>
>
> > Q6. Minor requested changes.
>
> A6. We have revised typos and thank the reviewer for pointing them out.
>
> [2] Junyu Shi, Wei Wan, Shengshan Hu, Jianrong Lu, and Leo Yu Zhang. Challenges and approaches for mitigating byzantine attacks in federated learning. In 2022 IEEE International Conference on Trust, Security and Privacy in Computing and Communications (TrustCom), 2022.

---

### Review · Reviewer_3KTv · 2023-12-14

**Summary Of Contributions:**

This paper studies the problem of robust federated learning (FL) in the presence of an informed training-time adversary. The idea is to formulate the problem as a mixed Nash problem where different robust aggregation rules and potential attacks compete with each other under a minimax framework to achieve robustness and better convergence. Certain assumptions were made to make it work, for example, the server knows the (estimate) of the true gradient update (knowing a subset of honest clients). A RobustTailor aggregation algorithm was proposed to solve the mixed Nash problem with good theoretical convergence guarantees. A set of experiments were conducted to valid the effectiveness of RobustTailor.

**Audience:**

Yes

**Broader Impact Concerns:**

No ethical concerns are identified for this paper.

**Claims And Evidence:**

Yes

**Requested Changes:**

- A clearer definition of the threat model and its difference to existing works. A summary/comparison table would be better.
- Why the methods are more stable under larger $\epsilon$.
- Add a detailed description of how Algorithm 1 works, i.e., the meaning/understanding of each step.
- Explain where the improvement comes from.
- Experiments with large models like ViT and high-resolution datasets like ImageNet (subset).
- Experiments against backdoor attacks like DBA.

**Strengths And Weaknesses:**

Strengths:
1. The study of an important problem, i.e., securing FL against training time adversaries, is a major vulnerability of FL.
2. The formulation of robust FL as a mixed Nash problem is interesting, and the proposed RobustTailor framework is novel.
3. Theoretical analysis is provided for the convergence of RobustTailor.

Weaknesses:
1. The assumptions are somewhat hard to follow. I.e., there seem to be multiple strong assumptions were made here and there, but is still not clear though: what is an informed adversary, why the server knows some honest clients, what is the goal of a Byzantine adversary, why backdoor attacks were not considered, and so on. And the differences of this threat model to other existing works.

2. The proposed RobustTailor method does not always outperform existing works like Krum or Comed. In Figure 1(a), Figure 9 (c) Comed works well at early training iterations but is very unstable at a later stage, but this seems to be fixable.  In Figure 1(b)/1(d), Figure 4(b), Figure 9(b) Krum works better,

3. Why RobustTailor work on unknown attacks? What about backdoor attacks, which are also one type of training time attack?

4. What is the difference between this work and GFA[1]? And why GFA is not compared in the experiments?

[1] Tahanian E, Amouei M, Fateh H, et al. A game-theoretic approach for robust federated learning[J]. International Journal of Engineering, 2021, 34(4): 832-842.

---

> ### Author Response · Authors · 2024-01-10
> **Response to Reviewer 3KTv (Part 1)**
>
> We thank the reviewer for their valuable feedback and address all remaining concerns below:
>
> > Q1. The assumptions are somewhat hard to follow. i.e., there seem to be multiple strong assumptions were made here and there, but is still not clear though: what is an informed adversary, why the server knows some honest clients, what is the goal of a Byzantine adversary, why backdoor attacks were not considered, and so on. And the differences of this threat model to other existing works.
>
> A1. For a clearer problem formulation, we revise Section 3 and list assumptions explictly (Assumption 1-3). For a better understanding, we also list the main assumptions in Section 3 here:
>
> ||Server| Adversary|
> |:---:|:---:|:---:|
> |Own strategy | $\mathcal{A} = \{\text{AG}_1, \ldots, \text{AG}_M\}$| $\mathcal{F} = \{\text{AT}_1, \ldots, \text{AT}_S\}$|
> |Opponent's strategy | $\mathcal{F}$ is known but the selection of $\text{AT}$ is unknown |$\mathcal{A}$ is known but the selection of $\text{AG}$ is unknown|
> |Honest gradients $\mathbf{g}^\star$ | Unknown | Known |
>
> We address the specific questions one by one below:
> - An informed adversary controls $f$ compromised clients and has access to all updates of $n-f$ honest clients.
> - The server doesn't know the honest clients. It only knows the upper bound on the number of compromised clients and can simulate them.
> - The adversary’s objective is to manipulate the global model at the server and minimize the test accuracy.
> - We specifically focus on tailored training-time attacks, categorized as poisoning availability attacks following [1], rather than backdoor attacks.
> - Our threat model is the same as existing literature [Fang et al., 2020; Xie et al., 2020a; Park et al., 2021; Ozfatura et al., 2022; Ramezani-Kebrya et al., 2022].
>
>
> > Q2. The proposed RobustTailor method does not always outperform existing works like Krum or Comed. In Figure 1(a), Figure 9(c). Comed works well at early training iterations but is very unstable at a later stage, but this seems to be fixable. In Figure 1(b)/1(d), Figure 4(b), Figure 9(b) Krum works better.
>
> A2. We emphasize that $\mathrm{RobustTailor}$ is a **framework** for aggregators rather than introducing a new aggregation rule. Its performance depends on the aggregators in the server's set. When defending against a single attack, $\mathrm{RobustTailor}$ can only achieve similiar performance with the best aggregator in the set rather than outperform it, especially if not all aggregation rules are immune to the specific attack Consequently, it is expected that Krum works better when against $\epsilon=100$. Moreover, instability in Comed can indeed signify susceptibility to large $\epsilon$ attacks, a factor also demonstrated in MixTailor [Ramezani-Kebrya et al., 2022].
>
>
> > Q3. Why RobustTailor work on unknown attacks? What about backdoor attacks, which are also one type of training time attack?
>
> A3. $\mathrm{RobustTailor}$, functioning as a framework comprising multiple aggregators, offers a higher likelihood of defending against unknown attacks compared to a single aggregator.
>
> Our focus primarily centers on tailored training-time attacks, specifically categorized as poisoning availability attacks following [1]. However, it's important to note that our study does not encompass poisoning integrity attacks as outlined in [1]. Regarding backdoor attacks, RobustTailor isn't explicitly designed to counter such attacks aiming to manipulate predictions for specific targeted data points. However, there is potential for $\mathrm{RobustTailor}$ to mitigate backdoor attacks if the server's set includes backdoor-resilient aggregators. This intriguing area remains a prospect for future exploration.
>
>
>
> > Q4. What is the difference between this work and GFA? And why GFA is not compared in the experiments?
>
> A4. GFA operates as an aggregation rule that constructs a game between the server and individual clients to identify malicious clients. However, $\mathrm{RobustTailor}$ is a **framework** capable of dynamically selecting suitable aggregators at each iteration. While GFA could indeed be included in the aggregator set, we did not incorporate it because our primary objective isn't to demonstrate that $\mathrm{RobustTailor}$, when paired with two fundamental aggregators, outperforms all existing aggregation rules. Our empirical findings illustrate that $\mathrm{RobustTailor}$ demonstrates the ability to adeptly select an appropriate aggregator during training, showcasing its expandablity across a wide range of aggregation rules.
>
>
> [1] Ambra Demontis, Marco Melis, Maura Pintor, Matthew Jagielski, Battista Biggio, Alina Oprea, Cristina Nita-Rotaru, and Fabio Roli. Why do adversarial attacks transfer? Explaining transferability of evasion and poisoning attacks. In Proc. USENIX security symposium, 2019.

---

> ### Author Response · Authors · 2024-01-10
> **Response to Reviewer 3KTv (Part 2)**
>
> > Q5. Other requested changes.
>
> A5. We briefly answer the requested changes except of those answered above one by one here:
> - We kindly disagree that the methods are more stable under larger $\epsilon$. It is only that the behaviors of Krum and Comed are different when attacked. Krum shows low accuracy from the beginning when facing small $\epsilon$ attacks while Comed shows fluctuations at the end of training.
> - We have added a detailed description of Algorithm 1 in Section 4.
> - The improvement of $\mathrm{RobustTailor}$ is that it is a framework boosting existing aggregation rules.
> - For ViT model on ImageNet, we claim that we follow existing work that also focuses on MNIST [Fang et al., 2020; Ramezani-Kebrya et al., 2022; Blanchard et al., 2017; Karimireddy et al., 2022] – note that this has been already challenging under Byzantine attacks. Furthermore, it's worth noting that none of these baseline papers provide empirical results on ImageNet. Scaling the experiments to ImageNet is interesting, but would be a separate contribution in its own right.

---

### Review · Reviewer_sisA · 2023-12-19

**Summary Of Contributions:**

The authors propose being robust to an attack on federated learning where a subset of clients are compromised. The problem is formulated as a game where both players use a no-regret learning algorithm to converge to the NE. The gradients are simulated, as the defender does not know the true updates of compromised clients.

**Audience:**

Yes

**Claims And Evidence:**

Yes

**Requested Changes:**

Please revise 4.1 to be more detailed and precise.

**Strengths And Weaknesses:**

The paper is well-written with solid experiments.
I feel that the assumption of a public subset of the dataset hurts privacy; I am not sure some supporting papers is good enough for that claim. I do not completely get the second paragraph in 4.1 - would few trusted clients not results in estimations that are far from ground truth?
Recently there have been attacks on no-regret algorithm - would it be better to try those rather than the simple thing done in Section 6.2?

---

> ### Author Response · Authors · 2024-01-05
> **Kindly request for clarification: attacks on no-regret algorithms**
>
> Dear Reviewer sisA,
>
> Happy new year! Thanks for your comment. However, we seek clarification regarding the attacks on no-regret algorithms you mentioned. We find several papers [1-4] discussing related themes but they are not entirely relevant to our paper's focus. I was wondering if there are other works more closely related to the attacks on no-regret algorithms that you have in mind. We would appreciate if you can provide some references. Thank you for considering this request.
>
> Best regards,
>
> Authors
>
> [1] Ma, Yuzhe, and Zhijin Zhou. Adversarial Attacks on Adversarial Bandits. arXiv preprint arXiv:2301.12595, 2023.
>
> [2] Zhu, Banghua, Lun Wang, Qi Pang, Shuai Wang, Jiantao Jiao, Dawn Song, and Michael I. Jordan. Byzantine-robust federated learning with optimal statistical rates. In International Conference on Artificial Intelligence and Statistics, 2023.
>
> [3] Zhu, Banghua, Jiantao Jiao, and Jacob Steinhardt. Robust estimation via generalized quasi-gradients. Information and Inference: A Journal of the IMA 11, 2022.
>
> [4] Hopkins, Sam, Jerry Li, and Fred Zhang. Robust and heavy-tailed mean estimation made simple, via regret minimization. Advances in Neural Information Processing Systems, 2020.

---

> > ### Comment · Reviewer_sisA · 2024-01-08
> > **No paper in particular**
> >
> > I was not referring to any paper in particular, but just asking to consider if any attacks on adversarial bandit are relevant here. Can the adversary control the loss? A discussion of why not relevant would also suffice.

---

> ### Author Response · Authors · 2024-01-10
> **Response to Reviewer sisA**
>
> We thank the reviewer for their valuable feedback and address all remaining concerns below:
>
>
> > Q1. I feel that the assumption of a public subset of the dataset hurts privacy. I am not sure some supporting papers is good enough for that claim.
>
> A1. While it is true that having a small public dataset may have a minor impact on privacy, it’s important to recognize that this is a deliberate trade-off between enhancing robustness and preserving privacy. In scenarios where the server is confronted with a knowledgeable adversary who possesses information about all honest clients and their updates, the server’s optimal course of action, in the absence of additional information, would be to randomly select one aggregator from the available set. However, $\mathrm{RobustTailor}$ improves upon this random selection by leveraging the additional information at the server’s disposal.
> Furthermore, it is worth noting that the concept of a public dataset has gained widespread acceptance within the field of FL [Kairouz et al., 2021; Fang and Ye, 2022; Huang et al., 2022; Yoshida et al., 2020], as highlighted in our discussion of Related Work. Additionally, we make the assertion that some companies rely on testers with a high level of mutual trust, as detailed in Section 4.1. Moreover, the empirical results presented in Section 6.4 clearly indicate that the public dataset’s distribution need only be a rough approximation of the true data distribution. Therefore, we maintain that the inclusion of a public dataset does not significantly compromise privacy and is indeed an acceptable compromise given the benefits it provides.
>
> > Q2. I do not completely get the second paragraph in 4.1 - would few trusted clients not results in estimations that are far from ground truth?
>
> A2. We note that there exists a trade-off between privacy and convergence guarantees. In particular, by decreasing $\epsilon,\delta$, stronger privacy guarantees are achieved for trusted clients; while it slows down convergence in Theorem 1 due to an increased $\text{V}_{\text{est}}$. In the revised version, we revise Section 4.1 and elaborate on this trade-off in Remark 2 in Appendix I.
>
> > Q3. Recently there have been attacks on no-regret algorithm - would it be better to try those rather than the simple thing done in Section 6.2?
>
> A3. We claim that $\mathrm{AttackTailor}$ we propose is a kind of adversarial bandit attack, which uses losses in the simulated game as rewards to choose proper attacks. Other adversarial bandit attacks on no-regret algorithm can be left for future work.

---

> > ### Comment · Reviewer_sisA · 2024-01-13
> > **Wording Section 4.1**
> >
> > I would have preferred to explicitly have the word "trade-off" as mentioned in your comment "deliberate trade-off between enhancing robustness and preserving privacy" - I do not see this in text, from text there is a sense of no privacy risk?

---

> > > ### Author Response · Authors · 2024-01-14
> > > **Revised the paper**
> > >
> > > Dear Reviewer sisA,
> > >
> > > Thank you for your valuable suggestion. We have addressed your concerns by revising Section 4.1, explicitly noting in the first sentence that this is a deliberate trade-off between enhancing robustness and preserving privacy. In addition, we have updated the Abstract to mention a "minor trade-off of privacy".
> > >
> > > Best regards,
> > >
> > > Authors

---

### Decision · Action_Editor_TywW · 2024-02-09

**Recommendation:** Accept as is

**Comment:**

This paper proposes a game-theoretical framework to enhance the robustness of federated learning to malicious clients by combining multiple aggregation rules dynamically. The proposed approach is validated by both theoretical analysis and numerical experiments. Overall, it is an interesting addition to the literature on robustness in federated learning, which has mostly considered the use of a single robust aggregation rule.

**Audience:**

Robustness in (federated) machine learning is of broad interest to TMLR's audience.

**Claims And Evidence:**

The claim that the proposed approach improves robustness to training-time attacks compared to using a single robust aggregation rule is supported by theoretical results and numerical experiments. Following the discussion with reviewers, the authors have slightly toned down a bit the claim that their approach does not harm privacy by acknowledging the impact of assuming the availability of "public" data.